# MASSIVELY SCALABLE INVERSE REINFORCEMENT LEARNING IN GOOGLE MAPS

**Matt Barnes**[1][*]   **Matthew Abueg**[1][*]   **Oliver F. Lange**[2]   **Matt Deeds**[2]
**Jason Trader**[2]   **Denali Molitor**[1]   **Markus Wulfmeier**[3][†]   **Shawn O'Banion**[1][†]
[1]Google Research   [2]Google Maps   [3]Google DeepMind

## ABSTRACT

Inverse reinforcement learning (IRL) offers a powerful and general framework for learning humans' latent preferences in route recommendation, yet no approach has successfully addressed planetary-scale problems with hundreds of millions of states and demonstration trajectories. In this paper, we introduce scaling techniques based on graph compression, spatial parallelization, and improved initialization conditions inspired by a connection to eigenvector algorithms. We revisit classic IRL methods in the routing context, and make the key observation that there exists a trade-off between the use of cheap, deterministic planners and expensive yet robust stochastic policies. This insight is leveraged in Receding Horizon Inverse Planning (RHIP), a new generalization of classic IRL algorithms that provides fine-grained control over performance trade-offs via its planning horizon. Our contributions culminate in a policy that achieves a 16-24% improvement in route quality at a global scale, and to the best of our knowledge, represents the largest published study of IRL algorithms in a real-world setting to date. We conclude by conducting an ablation study of key components, presenting negative results from alternative eigenvalue solvers, and identifying opportunities to further improve scalability via IRL-specific batching strategies.

## 1 INTRODUCTION

Inverse reinforcement learning (IRL) is the problem of learning latent preferences from observed sequential decision making behavior. First proposed by Rudolf Kálmán in 1964 (when it went under the name of inverse optimal control (Kalman, 1964), and later structural estimation (Rust, 1994)), IRL has now been studied in robotics (Abbeel et al., 2008; Ratliff et al., 2009; Ratliff et al., 2007), cognitive science (Baker et al., 2009), video games (Tastan and Sukthankar, 2011; Tucker et al., 2018), human motion behavior (Kitani et al., 2012; Rhinehart and Kitani, 2020) and healthcare (Imani and Braga-Neto, 2019; Yu et al., 2019), among others.

In this paper, we address a key challenge in all these applications: scalability (Chan and Schaar, 2021; Michini et al., 2013; Wulfmeier et al., 2016b). With several notable exceptions, IRL algorithms require solving an RL problem at every gradient step, in addition to performing standard backpropagation (Finn et al., 2016b; Swamy et al., 2023). This is a significant computational challenge, and necessitates access to both an interactive MDP and a dataset of expert demonstrations that are often costly to collect. By addressing the scalability issue, we aim to leverage recent advancements in training foundation-sized models on large datasets.

To illustrate our claims, we focus on the classic route finding task, due to its immediate practical significance and the availability of large demonstration datasets. Given an origin and destination location anywhere in the world, the goal is to provide routes that best reflect travelers' latent preferences. These preferences are only observed through their physical behavior, which implicitly trade-off factors including traffic conditions, distance, hills, safety, scenery, road conditions, etc. Although we primarily focus on route finding, the advancements in this paper are general enough to find use more broadly.

We address the scalability challenge by providing both (a) practical techniques to improve IRL scalability and (b) a new view on classic IRL algorithms that reveals a novel generalization and enables fine-grained control of performance characteristics. Concretely, our contributions are as follows:

---

[*][†]Equal contribution. Correspondence to {`mattbarnes,mattea,mwulfmeier,obanion`}`@google.com`

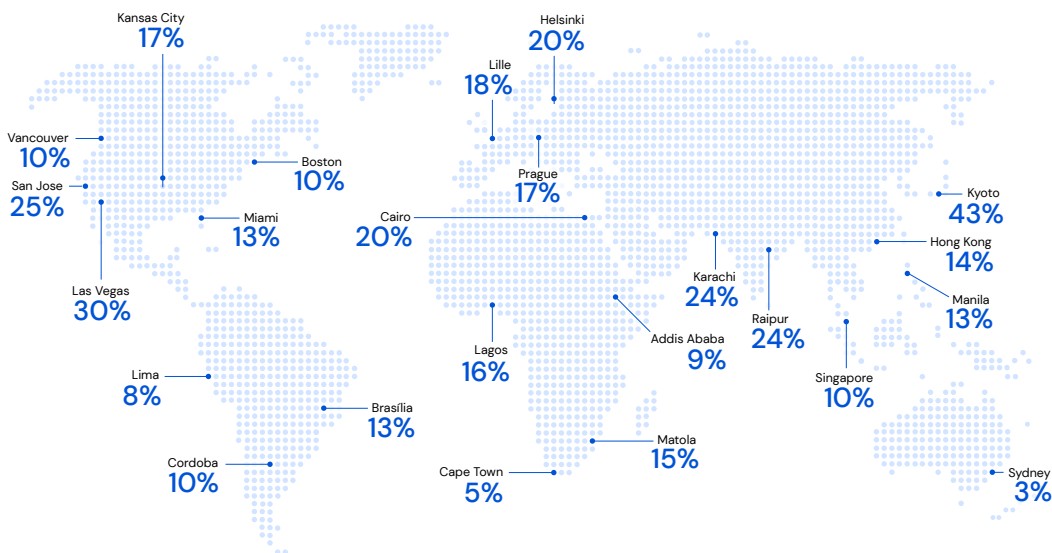

Figure 1: Google Maps route accuracy improvements in several world regions, when using our inverse reinforcement learning policy. Full results are presented in Table 1 and Figure 7.

- **MaxEnt++** An improved version of MaxEnt IRL (Ziebart et al., 2008) that is inspired by a connection to dominant eigenvectors to initialize the backward pass closer to the desired solution.

- **Spatial parallelization** A simple, practical technique to shard the global MDP by leveraging a geography-based sparse mixture-of-experts.

- **Graph compression strategies** Lossless and lossy methods to compress the graph matrices and reduce both the memory footprint and FLOP count across all IRL algorithms.

- **Receding Horizon Inverse Planning (RHIP)** A novel IRL algorithm that generalizes MaxEnt++, BIRL (Ramachandran and Amir, 2007) and MMP (Ratliff et al., 2006). RHIP leverages our insight that there exists a trade-off between the use of cheap, deterministic planners and expensive yet robust stochastic policies. Practically, RHIP enables interpolating between classic algorithms to realize policies that are both fast and accurate.

- **Alternative solvers** Secondary negative results when attempting to use Arnoldi iteration and a closed-form matrix geometric series in MaxEnt, which we defer to Appendix A.

- **MaxEnt theory** Secondary theoretical analyses of MaxEnt, which we defer to Appendix B.

Our work culminates in a global policy that achieves a 15.9% and 24.1% lift in route accuracy for driving and two-wheelers, respectively (Figure 1), and was successfully applied to a large scale setting in Google Maps. To the best of our knowledge, this represents the largest published study of IRL methods in a real-world setting to date.

## 2 INVERSE REINFORCEMENT LEARNING

A Markov decision process (MDP) $\mathcal{M}$ is defined by a discrete set of states $\mathcal{S}$, actions $\mathcal{A}$, transition kernel $\mathcal{T}$ and reward function $r$ (i.e. negative cost function). Given $\mathcal{M} \setminus r$ and a set of state-action trajectory demonstrations $\mathcal{D} = \{\tau_1, ..., \tau_N\}$ sampled from a demonstration policy $\pi_E$, the goal of IRL is to recover the latent $r$.[1] We denote the state and state-action distributions of trajectory $\tau$ by $\rho_\tau^s$ and $\rho_\tau^{sa}$, respectively.

For expository purposes, we initially restrict our attention to the classic path-planning problem, and discuss extensions in Section 6. In line with prior work (Ziebart et al., 2008), we define these MDPs as discrete, deterministic, and undiscounted where $r_\theta(s_i, s_j) \leq 0$ denotes the $\theta$-parameterized reward of transitioning from state $s_i$ to state $s_j$ and non-allowable transitions have reward $-\infty$. There exists a single self-absorbing zero-reward destination state $s_d$, which implies that each unique destination

---

[1]This is an ill-conditioned problem, as multiple reward functions can induce the same trajectories. Methods impose regularization or constraints to form a unique solution, e.g. the principle of maximum entropy (Ziebart et al., 2008).

induces a slightly different MDP. However, for the sake of notational simplicity and without loss of generality, we consider the special case of a single origin state $s_o$ and single destination state $s_d$. We do not make this simplification for any of our empirical results. In the path-planning context, states (i.e. nodes) represent road segments and allowable transitions (i.e. edges) between nodes represent turns.

Inverse reinforcement learning algorithms follow the two-player zero-sum game

$$\min_{\pi \in \Pi} \max_{\theta \in \Theta} J(\pi_E, r_\theta) - J(\pi, r_\theta) = \min_{\pi \in \Pi} \max_{\theta \in \Theta} f(\pi_E, \pi, r_\theta) \tag{1}$$

where $J(\pi, r_\theta) = \mathbb{E}_{\tau \sim \pi} \sum_{(s_t, s_{t+1}) \in \tau} r_\theta(s_t, s_{t+1})$ denotes the value of the policy $\pi$ under reward function $r_\theta$. We consider primal strategies for the equilibrium computation, where the policy player follows a no-regret strategy against a best-response discriminative player (Swamy et al., 2021). Classic IRL algorithms follow from Equation 1 by specifying certain policy classes and regularizers (see Appendix C for details). Following Ziebart et al. (2008), we refer to the *backward pass* as estimating the current policy $\pi$ and the *forward pass* as rolling out the current policy.

**Goal conditioning** Learning a function $r_\theta$ using IRL provides a concise representation of preferences and simplifies transfer across goal states $s_d$, as the reward function is decomposed into a general learned term and a fixed modification at the destination (self-absorbing, zero-reward). In the tabular setting, the number of reward parameters is $\mathcal{O}(SA)$ even when conditioning on $s_d$. This is in contrast to approaches that explicitly learn a policy, Q-function or value function (e.g. BC, IQ-Learn (Garg et al., 2021), ValueDICE (Kostrikov et al., 2020), GAIL (Ho and Ermon, 2016), DAGGER (Ross and D. Bagnell, 2010)), which require additional complexity when conditioning on $s_d$, e.g. in the tabular setting, the number of policy parameters increases from $\mathcal{O}(\mathcal{S A})$ to $\mathcal{O}(S^2 A)$. By learning rewards instead of policies, we can evaluate $r_\theta$ *once offline* for every edge in the graph, store the results in a database, precompute contraction hierarchies (Geisberger et al., 2012), and use a fast graph search algorithm to find the highest reward path[2] for online $(s_o, s_d)$ requests. This is in contrast to a learned policy, which must be evaluated *online for every $(s_o, s_d)$ request* and for every step in the sampled route – a computationally untenable solution in many online environments.

## 3 RELATED WORK

IRL approaches can be categorized according to the form of their loss function.[3] MaxEnt (Ziebart et al., 2008) optimizes cross-entropy loss, MMP (Ratliff et al., 2006) optimizes margin loss, and BIRL (Ramachandran and Amir, 2007) optimizes sequential Bayesian likelihood. LEARCH (Ratliff et al., 2009) replaces the quadratic programming optimization in MMP (Ratliff et al., 2006) with stochastic gradient descent. Choi and Kim (2011) replace the MCMC sampling in BIRL (Ramachandran and Amir, 2007) with maximum a posteriori estimation. Extensions to continuous state-action spaces are possible through sampling-based techniques (Finn et al., 2016a; Fu et al., 2018). Our work builds on Wulfmeier et al. (2015) and Mainprice et al. (2016), who applied MaxEnt and LEARCH to the deep function approximator setting.

Existing approaches to scale IRL consider several orthogonal and often complimentary techniques. Michini et al. (2013) incorporate real-time dynamic programming (Barto et al., 1995), which is less applicable with modern accelerators' matrix operation parallelization. Chan and Schaar (2021) apply a variational perspective to BIRL. MMP is inherently more scalable, as its inner loop only requires calling a planning subroutine (e.g. Dijkstra) (Ratliff et al., 2009). However, it lacks robustness to real-world noise, and has lost favor to more stable and accurate probabilistic policies (Osa et al., 2018). MacGlashan and Littman (2015) similarly introduce a receding horizon to improve planning times.[4] They assume zero reward beyond the horizon, whereas we assume a cheap deterministic planner beyond the horizon in order to propagate rewards from distant goal states.

Recently, imitation learning approaches that directly attempt to recover the demonstrator policy have gained increased attention (Ho and Ermon, 2016; Ke et al., 2021). Behavior cloning avoids performing potentially expensive environment roll-outs, but suffers regret quadratic in the horizon (Ross and D. Bagnell, 2010). DAGGER solves the compounding errors problem by utilizing expert corrections

---

[2]The highest reward path is equivalent to the *most likely* path under MaxEnt (Ziebart et al., 2008).

[3]The IRL route optimization problem is reducible to supervised classification with infinitely many classes, where each class is a valid route from the origin to the destination, the features are specified by the edges, and the label is the demonstration route. Unlike typical supervised learning problems, solving this directly by enumerating all classes is intractable, so IRL approaches take advantage of the MDP structure to efficiently compute loss gradients.

[4]MacGlashan and Littman (2015) reduces to BIRL (Ramachandran and Amir, 2007) with infinite horizon.

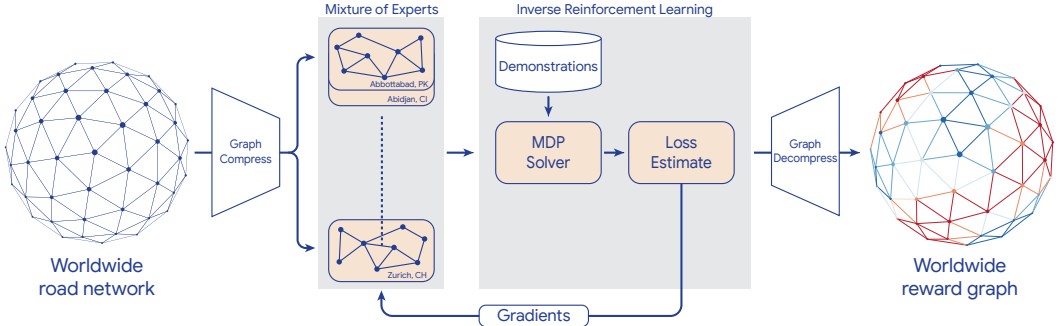

Figure 2: Architecture overview. The final rewards are used to serve online routing requests.

(Ross et al., 2011). Recent approaches use the demonstrators' distribution to reduce exploration in the RL subroutine (Swamy et al., 2023) and are complementary to our work. Mixed approaches such as GAIL simultaneously learn both a policy (generator) and reward function (discriminator) (Finn et al., 2016b; Ho and Ermon, 2016; Ke et al., 2021), although can be susceptible to training instabilities (Xing et al., 2021). IQ-Learn (Garg et al., 2021) and ValueDICE (Kostrikov et al., 2020) directly learn a Q-function and value function, respectively. We avoid explicitly learning a policy, Q-function or value function due to the goal conditioning requirement discussed in Section 2.

## 4 METHODS

The worldwide road network contains hundreds of millions of nodes and edges. At first glance, even attempting to fit the graph features into high-bandwidth memory to compute a single gradient step is infeasible. In this section, we present a series of advancements which enable solving the world-scale IRL route finding problem, summarized in Figure 2. At the end of Section 6, we provide a useful summary of other directions which yield negative results.

**Parallelism strategies** We use a sparse Mixture of Experts (MoE) strategy (Shazeer et al., 2017), where experts are uniquely associated to geographic regions and each demonstration sample is deterministically assigned to a single expert (i.e. one-hot sparsity). This minimizes cross-expert samples and allows each expert to learn routing preferences specific to its region. Specifically, we shard the global MDP $\mathcal{M}$ and demonstration dataset $\mathcal{D}$ into $m$ disjoint subproblems $(\mathcal{M}_1, \mathcal{D}_1),...,(\mathcal{M}_m, \mathcal{D}_m)$. We train region-specific experts $r_{\theta_1},...,r_{\theta_m}$ in parallel, and compute the final global rewards using $r(s,a) = r_{\theta_i}(s,a)$ where $(s,a) \in \mathcal{M}_i$. We additionally use standard data parallelism strategies within each expert to further partition minibatch samples across accelerator devices.

**MaxEnt++ initialization** MaxEnt (Ziebart et al., 2008) is typically initialized to the value (i.e. log partition) function $v^{(0)}(s_i) = \log \mathbb{I}[s_i = s_d]$, where $\mathbb{I}[s_i = s_d]$ is zero everywhere except at the destination node (see Appendix C.1). This propagates information outwards from the destination, and requires that the number of dynamic programming steps is at least the graph diameter for arbitrary destinations. Instead, we initialize the values $v^{(0)}$ to be the *highest reward to the destination from every node*. This initialization is strictly closer to the desired solution $v$ by

$$\underbrace{\mathbb{I}[s_i = s_d]}_{\text{MaxEnt initialization}} \leq \underbrace{\min_{\tau \in \mathcal{T}_{s_i,s_d}} e^{r(\tau)}}_{\text{MaxEnt++ initialization}} \leq \underbrace{\sum_{\tau \in \mathcal{T}_{s_i,s_d}} e^{r(\tau)} = v(s_i)}_{\text{Solution}} \tag{2}$$

where $\mathcal{T}_{s_i,s_d}$ is the (infinite) set of all paths which begin at $s_i$ and end at $s_d$ (see proof in Appendix B.3). Note that equality only holds on contrived MDPs, and the middle term can be cheaply computed via Dijkstra or A$^*$. We call this method MaxEnt++ (summarized in Algorithm 2).[5]

The correspondence between MaxEnt and power iteration provides a more intuitive perspective. Specifically, the MaxEnt backward pass initialization $e^{v^{(0)}}$ defines the initial conditions of power iteration, and the solution $e^v$ is the dominant eigenvector of the graph. By more closely aligning the initialization $v^{(0)}$ to the solution, the number of required power iteration steps is decreased.

---

[5]a nod to the improved initialization of k-means++ (Arthur and Vassilvitskii, 2006).

---

**Algorithm 1** RHIP (Receding Horizon Inverse Planning)

---

**Input:** Current reward $r_\theta$, horizon $H$,
demonstration $\tau$ with origin $s_o$ and destination $s_d$
**Output:** Parameter update $\nabla_\theta f$

```
# Policy estimation
```
$v^{(0)}(s) \leftarrow \text{DIJKSTRA}(r_\theta, s, s_d)$
$\pi_d(a|s) \leftarrow \text{Greedy}(r_\theta(s,a) + v^{(0)}(s'))$
**for** $h = 1, 2, ..., H$ **do**
$\quad Q^{(h)}(s,a) \leftarrow r_\theta(s,a) + v^{(h-1)}(s')$
$\quad v^{(h)}(s) \leftarrow \log\sum_a \exp Q^{(h)}(s,a)$
**end for**
$\pi_s(a|s) \leftarrow \frac{\exp Q^{(H)}(s,a)}{\sum_{a'} \exp Q^{(H)}(s,a')}$
$\pi \leftarrow \left[(\pi_s)_{\times H}, (\pi_d)_{\times\infty}\right]$ $\qquad \triangleright$ Equation 3

```
# Roll-out  πs → πd
```
$\rho_\theta^{sa} \leftarrow \text{Rollout}(\pi, \rho_\tau^s)$
$\rho_*^{sa} \leftarrow \text{Rollout}(\pi_{2:\infty}, \rho_{\tau 2:\infty}^s) + \rho_\tau^{sa}$

$\nabla_\theta f \leftarrow \sum_{s,a}(\rho_*^{sa}(s,a) - \rho_\theta^{sa}(s,a))\nabla_\theta r_\theta(s,a)$
**Return:** $\nabla_\theta f$

---

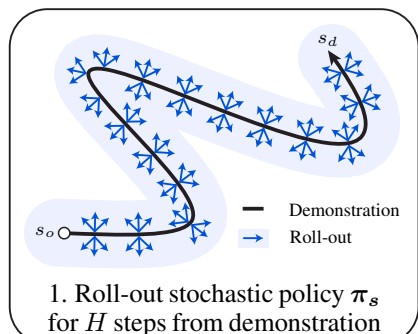

1. Roll-out stochastic policy $\pi_s$
for $H$ steps from demonstration

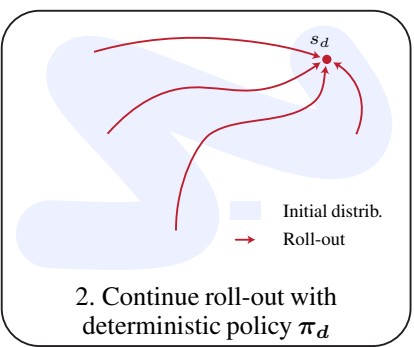

2. Continue roll-out with
deterministic policy $\pi_d$

Figure 3: RHIP rolls out the computationally expensive stochastic policy $\pi_s$ for $H$ steps from the demonstration path before transitioning to the cheaper deterministic policy $\pi_d$, as described in Equation 3. DIJKSTRA computes the highest reward path from state $s$ to $s_d$ under $r_\theta$. $\text{Greedy}(Q(s,a)) \in \{0,1\}$ is the deterministic policy that selects the action with the highest $Q$-value. $\text{Rollout}(\pi, \rho^s) = \rho^{sa}$ computes the state-action distribution $\rho^{sa}$ from rolling out the policy $\pi$ from the initial state distribution $\rho^s$ until convergence to the destination $s_d$.

**Receding Horizon Inverse Planning (RHIP)** In this section, our key insight is that classic IRL algorithms exhibit a trade-off between the use of cheap, deterministic planners (e.g. MMP (Ratliff et al., 2006)) and the use of expensive yet robust stochastic policies (e.g. MaxEnt (Ziebart et al., 2008)). This insight reveals a novel generalized algorithm that enables fine-grained control over performance characteristics and provides a new view on classic methods.

First, let $\pi_s$ denote the stochastic policy after $H$ steps of MaxEnt++ and $\pi_d$ denote the deterministic policy that follows highest reward path, i.e. $\pi_d(a|s) \in \{0,1\}$. Let $\mathcal{T}_{s,a}$ denote the set of all paths which begin with state-action pair $(s,a)$. We introduce a new policy defined by

$$\pi(a|s) \propto \sum_{\tau \in \mathcal{T}_{s,a}} \underbrace{\pi_d(\tau_{H+1})}_{\text{Deterministic policy}} \prod_{h=1}^{H} \underbrace{\pi_s(a_h|s_h)}_{\text{Stochastic policy}} . \tag{3}$$

The careful reader will notice that Equation 3 reduces to classic IRL algorithms for various choices of $H$. For $H = \infty$ it reduces to MaxEnt++, for $H = 1$ it reduces to BIRL (Ramachandran and Amir, 2007), and for $H = 0$ it reduces to MMP (Ratliff et al., 2006) with margin terms absorbed into $r_\theta$ (see Appendix C for details).

We call this generalization Receding Horizon Inverse Planning (RHIP, pronounced *rip*). As described in Algorithm 1, RHIP performs $H$ backup steps of MaxEnt++, rolls out the resulting stochastic policy $\pi_s$ for $H$ steps, and switches to rolling out the deterministic policy $\pi_d$ until reaching the destination. The receding horizon $H$ controls RHIP's compute budget by trading off the number of stochastic and deterministic steps. The stochastic policy $\pi_s$ is both expensive to estimate (backward pass) and roll-out (forward pass) compared to the deterministic policy $\pi_d$, which can be efficiently computed via Dijkstra's algorithm.

Table 1: Route quality of manually designed and IRL baselines. Due to the high computational cost of training the global model (bottom 3 rows), we also evaluate in a smaller, more computationally tractable set of metros (top section). Metrics are NLL (negative log-likelihood), Acc (accuracy, i.e. perfect route match) and IoU (Intersection over Union of trajectory edges). Numbers in bold are statistically significant with p-value less than .122 (see Appendix D.3). Two-wheeler data is unavailable globally.

| Policy class | Reward $r_\theta$ | Drive | | | Two wheelers | | |
|---|---|---|---|---|---|---|---|
| | | NLL | Acc | IoU | NLL | Acc | IoU |
| ETA | Linear | | .4034 | .6566 | | .4506 | .7050 |
| ETA+penalties | Linear | | .4274 | .6823 | | .4475 | .7146 |
| MMP/LEARCH [34, 35] | Linear | | .4244 | .6531 | | .4687 | .7054 |
| | SparseLin | | .4853 | .7069 | | .5233 | .7457 |
| Deep LEARCH [29] | DNN | | .4241 | .6532 | | .4777 | .7141 |
| | DNN+SparseLin | | .4682 | .6781 | | .5220 | .7300 |
| BIRL [9, 32] | Linear | 3.933 | .4524 | .6945 | 3.629 | .4933 | .7314 |
| | SparseLin | 26.840 | .4900 | .7084 | 8.975 | .5375 | .7508 |
| Deep BIRL | DNN | 3.621 | .4617 | .6958 | 3.308 | .4973 | .7340 |
| | DNN+SparseLin | 2.970 | .4988 | .7063 | 2.689 | .5546 | .7587 |
| MaxEnt [53], MaxEnt++ | Linear | 4.4409 | .4521 | .6941 | 3.957 | .4914 | .7293 |
| | SparseLin | 26.749 | .4922 | .7092 | 8.876 | .5401 | .7522 |
| Deep MaxEnt [46] | DNN | 3.729 | .4544 | .6864 | 3.493 | .4961 | .7308 |
| | DNN+SparseLin | 2.889 | .5007 | .7062 | 2.920 | .5490 | .7516 |
| RHIP | Linear | 3.930 | .4552 | .6965 | 3.630 | .4943 | .7319 |
| | SparseLin | 26.748 | .4926 | .7095 | 8.865 | .5408 | .7522 |
| | DNN | 3.590 | .4626 | .6955 | 3.295 | .5000 | .7343 |
| | DNN+SparseLin | 2.881 | **.5030** | .7086 | 2.661 | **.5564** | .7591 |
| Global ETA | Linear | | .3891 | .6538 | | | |
| Global ETA+penalties | Linear | | .4283 | .6907 | | | |
| Global RHIP | DNN+SparseLin | 8.194 | **.4958** | **.7208** | | | |

**Graph compression**    We introduce two graph compression techniques to reduce both the memory footprint and FLOP count across all IRL algorithms. The graph adjacency matrix is represented by a $B \times S \times V$ tensor, where entry $(b,s,v)$ contains the reward of the $v$'th edge emanating from node $s$ in batch sample $b$. Thus, $V$ is the maximum node degree valency, and nodes with fewer than $V$ outgoing edges are padded. For many problems, $V$ is tightly bounded, e.g. typically $V < 10$ in road networks. First, we perform lossless compression by 'spliting' nodes with degree close to $V$ into multiple nodes with lower degree. Since the majority of nodes have a much smaller degree than $V$, this slightly increases $S$ but can significantly decrease the effective $V$, thus reducing the overall tensor size $BSV$ in a lossless fashion. Second, we perform lossy compression by 'merging' nodes with a single outgoing edge into its downstream node as there is only one feasible action. Feature vectors of the merged nodes are summed, which is lossless for linear $r_\theta$ but introduces approximation error in the nonlinear setting. Intuitively, these compression techniques balance the graph's node degree distribution to reduce both the tensor padding (memory) and FLOP counts.

## 5    EMPIRICAL STUDY

**Road graph**    Our 200M state MDP is created from the Google Maps road network graph. Edge features contain predicted travel duration (estimated from historical traffic) and other relevant static road properties, including distance, surface condition, speed limit, name changes and road type.

**Demonstration dataset**    Dataset $\mathcal{D}$ contains de-identified users' trips collected during active navigation mode (Google, 2024). We filter for data quality by removing trips which contain loops, have poor GPS quality, or are unusually long. The dataset is a fixed-size subsample of these routes, spanning a period of two weeks and evenly split into training and evaluation sets based on date. Separate datasets are created for driving and two-wheelers, with the two-wheeler (e.g. mopeds, scooters) dataset being significantly smaller than the drive dataset due to a smaller region where this feature is available. The total number of iterated training and validation demonstration routes are 110M and 10M, respectively. See Appendix D for details.

**Preferred route**     **Detour route**

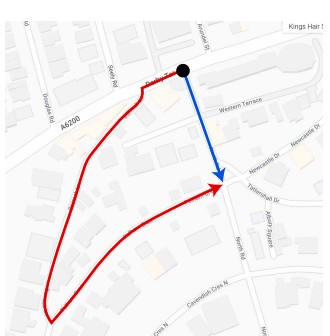 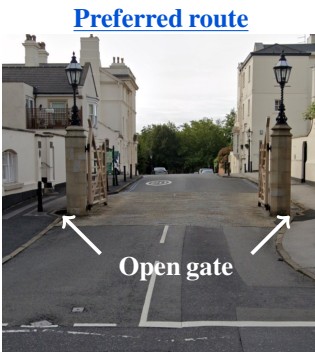 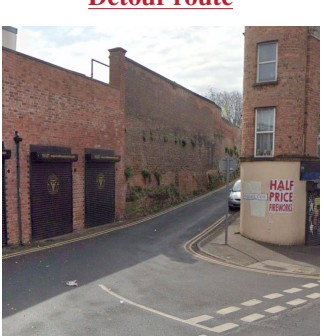

Open gate

Figure 4: Example of the 360M parameter sparse model finding and correcting a data quality error in Nottingham. The preferred route is incorrectly marked as private property due to the presence of a gate (which is never closed), and incorrectly incurs a high cost. The detour route is long and narrow. The sparse model learns to correct the data error with a large positive reward on the gated segment. Additional examples are provided in Appendix D.

**Experimental region**     Due to the high computational cost of training the global model, we perform initial hyperparameter selection on a smaller set of 9 experimental metros (Bekasi, Cairo, Cologne, Kolkata, Manchester, Manila, Nottingham, Orlando, and Syracuse). The top-performing configuration is used to train the global driving model. Two-wheeler data is unavailable globally and thus not reported.

**Baselines**     We evaluate both manually designed and IRL baselines. For fixed baselines, we consider (1) `ETA`: The fastest route, i.e. edge costs are the predicted travel duration and (2) `ETA+penalties`: ETA plus manually-tuned penalties for intuitively undesirable qualities (e.g. u-turns, unpaved roads), delivered to us in a closed form without visibility into the full set of underlying features. For IRL policy baselines, we compare MaxEnt (Algorithm 2) (Ziebart et al., 2008), Deep MaxEnt (Wulfmeier et al., 2015), the LEARCH (Ratliff et al., 2009) variation of MMP (Ratliff et al., 2006) (Algorithm 4), Deep LEARCH (Mainprice et al., 2016), and the maximum a posteriori variation of BIRL (Choi and Kim, 2011) (Algorithm 3). We also consider a deep version of BIRL similar to Brown and Niekum (2019).

**Reward model descriptions**     We evaluate three MoE function approximator classes: (1) a simple linear model, (2) a dense neural network (DNN) and (3) an $\ell_1$-regularized reward parameter for every edge in the graph (SparseLin). The latter is of particular interest because it tends to highlight data-quality issues, for example in Figure 4. These models have 3.9k, 144k, and 360M global parameters, respectively. We constrain model weights to produce non-positive rewards and fine-tune all models from the `ETA+penalties` baseline. DNN+SparseLin indicates additive DNN and SparseLin components. See Appendix D.2 for details.

**Metrics**     For serving online routing requests, we are interested in the highest reward path from $s_o$ to $s_d$ path under $r_\theta$ (and not a probabilistic sample from $\pi$ or a margin-augmented highest reward path). For accuracy, a route is considered correct if it perfectly matches the demonstration route. Intersection over Union (IoU) captures the amount of overlap with the demonstration route, and is computed based on unique edge ids. Negative log-likelihood (NLL) loss is reported where applicable.

## 5.1 RESULTS

We train the final global policy for 1.4 GPU-years on a large cluster of V100 machines, which results in a significant 15.9% and 24.1% increase in route accuracy relative to the `ETA+penalties` baseline models for driving and two-wheelers, respectively. As shown in Table 1, this RHIP policy with the largest 360M parameter reward model achieves state-of-the-art results with statistically significant accuracy gains of 0.4% and 0.2% compared to the next-best driving and two-wheeler IRL policies, respectively.

We observe dynamic programming convergence issues and large loss spikes in MaxEnt which tend to occur when the rewards become close to zero. In Appendix B we prove this phenomena occurs precisely when the dominant eigenvalue of the graph drops below a critical threshold of 1 (briefly noted in Ziebart (2010, pg. 117)) and show the set of allowable $\theta$ is provably convex in the linear case. Fortunately, we are able to manage the issue with careful initialization, learning rates, and stopping conditions. Note

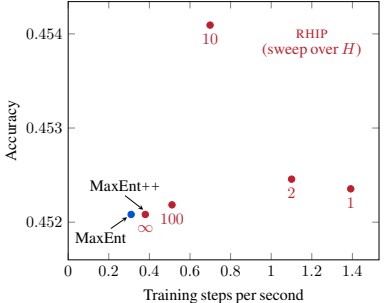

| Graph compression | $N$ | $V$ | Steps per sec | NLL | Acc |
|---|---|---|---|---|---|
| None | 124,402 | 4.9 | .373 | 9.371 | .454 |
| Split | 125,278 | 3.0 | .412 | 9.376 | .455 |
| Merge | 84,069 | 4.9 | .843 | 9.381 | .455 |
| Split+Merge | 84,944 | 3.0 | **.993** | 9.389 | .455 |

Figure 5: Impact of the horizon on accuracy and training time. $H=10$ has the highest accuracy and trains 70% faster than MaxEnt.

Table 2: Graph compression improves training time by 2.7x with insignificant impact on route quality metrics.

that RHIP (for $H < \infty$), BIRL and MMP provably do not suffer from this issue. All value functions (Algorithms 1, 2 and 3) are computed in log-space to avoid significant numerical stability issues.

We empirically study the trade-off between the use of cheap, deterministic planners and more robust yet expensive stochastic policies in Figure 5. As expected, MaxEnt has high accuracy but is slow to train due to expensive dynamic programming, and MaxEnt++ is 16% faster with no drop in accuracy. RHIP enables realizing a broad set of policies via the choice of $H$. Interestingly, we find that $H=10$ provides both the best quality routes and 70% faster training times than MaxEnt, i.e. MaxEnt is not on the Pareto front. We hypothesize this occurs due to improved policy specification. BIRL and MaxEnt assume humans probabilistically select actions according to the highest reward path or reward of all paths beginning with the respective state-action pair, respectively. However, in practice, humans may take a mixed approach – considering all paths within some horizon, and making approximations beyond that horizon.

Table 2 shows the impact of graph compression in our experimental region. The `split` strategy is lossless (as expected), and the `split+merge` strategy provides a significant 2.7x speed-up with almost no impact on route quality metrics. All empirical results take advantage of the `split+merge` graph compression. We find that data structure choice has a significant impact on training time. We try using unpadded, coordinate format (COO) sparse tensors to represent the graph adjacency matrix, but profiling results in our initial test metro of Bekasi show it to be 50x slower.

In Figure 6, we study the local geographic preferences learned by each expert in the mixture by performing an out-of-region generalization test. The drop in off-diagonal performance indicates the relative significance of local preferences. In Figure 7, we examine the relationship between region size and the performance of the model. Accuracy is nearly constant with respect to the number of states. However, training is significantly faster with fewer states, implying more equally sized regions would improve computational load balancing.

**Negative results**   We study several other ideas which, unlike the above contributions, do not meaningfully improve scalability. First, the MaxEnt backward pass is equivalent to applying power iteration to solve for the dominant eigenvector of the graph (Equation 4). Instead of using power iteration (Algorithm 2), we consider using Arnoldi iteration from ARPACK (Lehoucq et al., 1998), but find it to be numerically unstable due to lacking a log-space implementation (see results in Appendix A.1). Second, the forward pass used in MaxEnt has a closed form solution via the matrix geometric series (Equation 6). Using UMFPACK (Davis, 2004) to solve for this solution directly is faster on smaller graphs containing up to around 10k nodes, but provides no benefit on larger graphs (see results in Appendix A.2).

## 6   DISCUSSION

**Future research**   We study multiple techniques for improving the scalability of IRL and believe there exist several future directions which could lead to additional gains. First, demonstration paths with the same (or nearly the same) destination can be merged together into a single sample. Instead of performing one IRL backward and forward pass per mini-batch sample, we can now perform one iteration *per destination*. This enables compressing large datasets and increasing the effective batch size, although it reduces shuffling quality, i.e. samples are correlated during training. Second, the graph could be pruned by removing nodes not along a 'reasonable' path (i.e. within some margin of

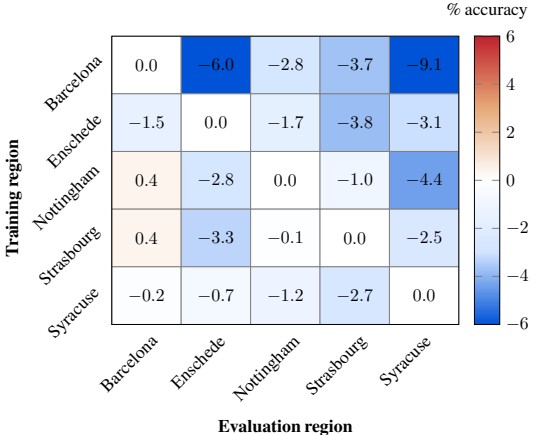 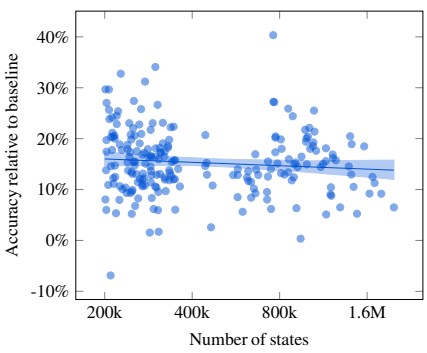

Figure 6: Sparse mixture-of-experts learn preferences specific to their geographic region, as demonstrated by the drop in off-diagonal performance.

Figure 7: Region accuracy for each expert in the worldwide mixture. Performance is consistent across the state space size.

the current best path) from $s_o$ to $s_d$ (Ziebart, 2010, pg. 119). However, this may be difficult to do in practice since the pruning is unique for every origin-destination pair.

From an accuracy perspective, we find that the sparse mixture-of-experts learn routing preferences specific to geographic regions. This vein could be further pursued with personalization, potentially via a hierarchical model (Choi and Kim, 2014; Choi and Kim, 2012). Due to engineering constraints, we use static ETA predictions, but would like to incorporate the dynamic GraphNet ETA predictions from Derrow-Pinion et al. (2021). We find that sparse reward models tend to highlight groups of edges which were impacted by the same underlying data quality issue. A group lasso penalty may be able to leverage this insight. Including human domain knowledge may robustify and help shape the reward function (Wulfmeier et al., 2016a; Wulfmeier et al., 2017). In this paper, we evaluate performance on driving and two-wheelers, and would like to incorporate other modes of transportation – especially walking and cycling – but are limited by engineering constraints.

**Extensions to other MDPs**   For expository purposes, we restrict attention to discrete, deterministic, and undiscounted MDPs with a single self-absorbing destination state. Our contributions naturally extend to other settings. MaxEnt++ and RHIP can be applied to any MDP where MaxEnt is appropriate and $v^{(0)}$ can be (efficiently) computed, e.g. via Dijkstra's. Our parallelization extends to all MDPs with a reasonable partition strategy, and the graph compression extends to stochastic MDPs (and with further approximation, discounted MDPs).

**Limitations**   IRL is limited by the quality of the demonstration routes. Even with significant effort to remove noisy and sub-optimal routes from $\mathcal{D}$, our policy will inadvertently learn some rewards which do not reflect users' true latent preferences. Our MoE strategy is based on *geographic regions*, which limits the sharing of information across large areas. This could be addressed with the addition of global model parameters. However, the abundance of demonstrations and lack of correlation between region size and accuracy (Figure 7) suggests benefits may be minimal.

## 7   CONCLUSION

Increasing performance via increased scale – both in terms of dataset size and model complexity – has proven to be a persistent trend in machine learning. Similar gains for inverse reinforcement learning problems have historically remained elusive, largely due to additional challenges posed by scaling the MDP solver. We contribute both (a) techniques for scaling IRL to problems with hundreds of millions of states, demonstration trajectories, and reward parameters and (b) a novel generalization that enables interpolating between classic IRL methods and provides fine-grained control over accuracy and planning time trade-offs. Our final policy is applied in a large scale setting with hundreds of millions of states and demonstration trajectories, which to the best of our knowledge represents the largest published study of IRL algorithms in a real-world setting to date.

ACKNOWLEDGMENTS

We gratefully acknowledge the contributions of Renaud Hartert, Rui Song, Thomas Sharp, Rémi Robert, Zoltan Szego, Beth Luan, Brit Larabee and Agnieszka Madurska towards an early exploration of this project for cyclists. Arno Eigenwillig provided useful suggestions for the graph's padded data structure, Jacob Moorman provided insightful discussions on the theoretical aspects of eigenvalue solvers, Jonathan Spencer provided helpful references for MaxEnt's theoretical analysis, and Ryan Epp explored the feasibility of an alternative approach to re-rank a small set of candidate routes. We are thankful for Remi Munos', Michael Bloesch's, Arun Ahuja's and the anonymous reviewers' feedback on final iterations of this work.

ETHICS STATEMENT

We strongly support open-sourcing experiments when legally and ethically permissible. The routing dataset used in this paper – and likely any routing dataset of similar scope – cannot be publicly released due to strict user privacy requirements.

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

# Appendix for "Massively Scalable Inverse Reinforcement Learning in Google Maps"

**Notation**   Let $R \in \mathbb{R}^{|\mathcal{S}| \times |\mathcal{S}|}$ be the sparse reward matrix defined by $R_{ij} = r_\theta(s_i, s_j)$.

## A   NEGATIVE RESULTS

In this section, we provide negative results for two ideas to improve MaxEnt's scalability. We used a synthetic Manhattan-style street grid (i.e. GridWorld) in order to conduct controlled sweeps over the number of nodes. Edge rewards were fixed at constant values.

### A.1   ARNOLDI ITERATION

The backward pass in MaxEnt is equivalent to performing power iteration to compute the dominant eigenvector of the graph

$$A_{ij} = e^{R_{ij}}. \tag{4}$$

We attempted to use Arnoldi iteration, as implemented in `ARPACK`, but found it to be numerically unstable due to lacking a log-space version (Lehoucq et al., 1998). In typical eigenvector applications, one is concerned with the reconstruction error $||Az - z||$, where $z = e^v$ is an eigenvector estimate of $A$. However, in MaxEnt we are uniquely concerned with relative error between entries in rows of $Q = R + \mathbb{1} \cdot v^\intercal$, since this determines the policy $\pi = \mathrm{softmax}(Q)$. We found log-space reconstruction error $||\log(Az) - \log(z)||$ to be far more significant in MaxEnt. As seen in Figure 8, Arnoldi iteration's reconstruction error in linear space is reasonable, but quickly blows up in log space, unlike log-space power iteration.

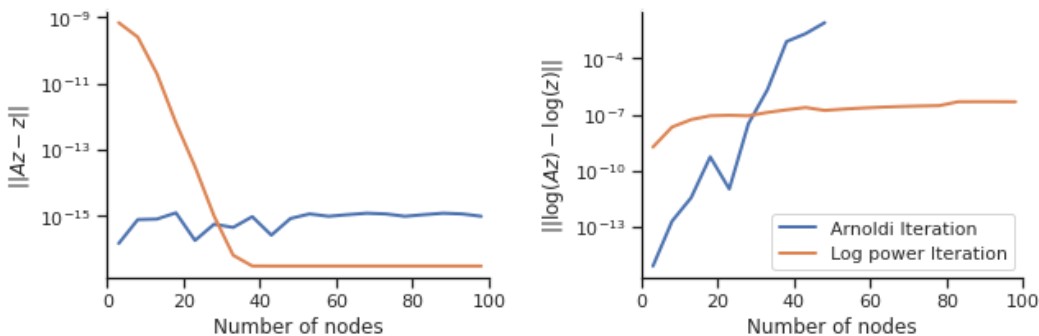

Figure 8: Solving for the dominant eigenvector in the MaxEnt backward pass.

From a geometric perspective, this phenomena can be observed by examining eigenvector estimates $z$ in Figure 9. Due to the MDP structure containing a single absorbing zero-reward destination node, eigenvector entries $z$ rapidly decay as one moves further away from the destination $s_d$. Thus, the typical reconstruction error $||Az - z||$ is primarily a function of a handful of nodes near the destination. Visually, Arnoldi and log-space power iteration appear to provide similar solutions for $z$. However, examining $\log(z)$ reveals that log-space power iteration is able to gracefully estimate the exponentially decaying eigenvector entries. Arnoldi iteration successfully estimates the handful of nodes near $s_d$, but further nodes are inaccurate and often invalid (white spaces indicate $z < 0$ and thus $\log(z)$ is undefined).

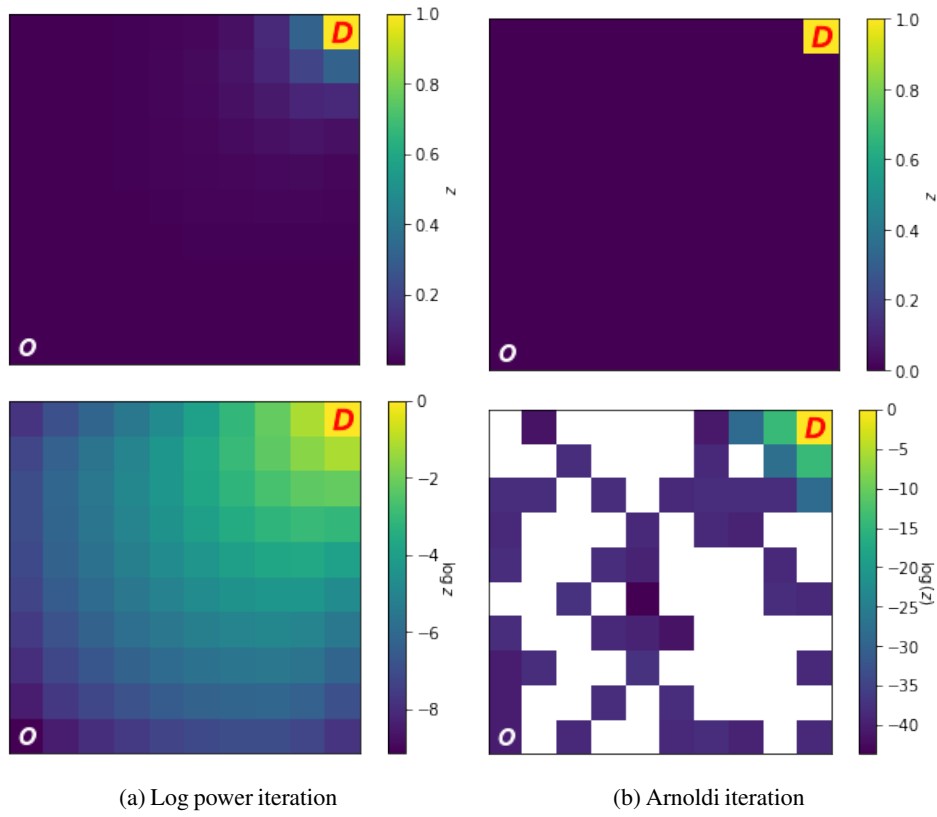

(a) Log power iteration        (b) Arnoldi iteration

Figure 9: Eigenvector estimates on a $10 \times 10$ GridWorld environment for origin O and destination D.

## A.2    MATRIX GEOMETRIC SERIES

The MaxEnt forward pass is a matrix geometric series with the closed-form solution defined in Equation 6. Instead of iteratively computing the series, we used `UMFPACK` to directly solve for this solution. As shown in Figure 10, this worked very well for graphs up to approximately 10k nodes, but provided no further benefit beyond this point. Neither approach had numerical stability issues.

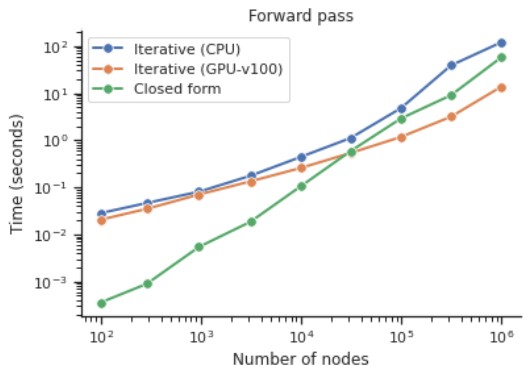

Figure 10: Closed-form solution of the MaxEnt forward pass using `UMFPACK`.

## A.3    NEGATIVE DESIGNS

We dismiss two other designs based on key pieces of evidence. First, we could learn to rerank the $\sim5$ routes returned by the Maps API (the approach taken by Ziebart et al. (2008)). This is a trivial solution to scale, but significantly reduces the route accuracy headroom. The API returns a handful of candidate routes from $s_o$ to $s_d$, which then form $\mathcal{T}$, e.g. Equation 2 or Equation 3. Instead of learning the best route

from a set of infinitely many routes, IRL now learns the best route among $\sim 5$. In many cases, the desired demonstration is often not among this set, and thus impossible to select in an online setting. Second, we could train in a smaller geographic region and deploy worldwide. Not only does this preclude the use of the best-performing 360M parameter sparse reward function (as its parameters are unique to each edge), Figure 6 indicates this would suffer from generalization error for other choices of $r_\theta$.

## B  MAXENT THEORY

In this section, we provide theoretical results on MaxEnt, which are summarized here for convenience:

- Theorem B.1: MaxEnt has finite loss if and only if the dominant eigenvalue $\lambda_{\max}$ of the graph is less than 1. This was briefly noted in Ziebart (2010, pg. 117)), which we discovered after preparation of this publication.
- Theorem B.2: The set of finite losses is convex in the linear setting.
- Theorem B.3: The convergence rate of the backward pass is governed by the second largest eigenvalue.

Figure 11 illustrates the first two phenomena on a simple didactic MDP. Intuitively, there are two counteracting properties which determine the transition between finite and infinite loss. First, the number of paths of length $n+1$ is larger than the number of paths of length $n$. This tends to accumulate probability mass on the set of longer paths. Second, paths of length $n+1$ have on average lower per-path reward, and thus lower per-path probability, than paths of length $n$. This favors the set of shorter paths. The question is, does the per-path probability decrease faster than the rate at which the number of paths increases? If yes, then the dominant eigenvalue is 1, the forward pass converges, and the loss is finite. If no, then the dominant eigenvalue is greater than 1, the forward pass never converges, and loss is infinite. As expected, infinite loss occurs in regions of high edge rewards, which agrees with Figure 11.

### B.1  PRELIMINARIES

Let $A_{ij} = e^{R_{ij}}$ be the elementwise exponential rewards. We restrict our attention to linear reward functions $R_{ij} = \theta^\mathsf{T} x_{ij}$ where $x_{ij}$ is set of edge features between nodes $i$ and $j$. If no edge exists between nodes $i$ and $j$, then the reward is negative infinite and $A_{ij} = 0$. Without loss of generality, we assume the destination node is the last row in $A$. Thus, $A$ is block upper triangular (also called the *irreducible normal form* or *Frobenius normal form*)

$$A = \begin{bmatrix} B_1 & * \\ 0 & B_2 \end{bmatrix}$$

where $B_1$ contains all the non-destination nodes and $B_2 = [1]$ is the self-absorbing destination node. $B_1$ and $B_2$ are both strongly connected (thus irreducible) and regular. $B_2$ has a single eigenvalue $\lambda_{\max}(B_2) = 1$, and by the Perron-Frobenius theorem $B_1$ has a dominant real eigenvalue $\lambda_{\max}(B_1)$ (called the Perron-Frobenius eigenvalue of $B_1$).

Likewise, we express an eigenvector $z$ of $A$ as

$$\begin{bmatrix} B_1 & * \\ 0 & B_2 \end{bmatrix} \begin{bmatrix} z_1 \\ z_2 \end{bmatrix} = \lambda \begin{bmatrix} z_1 \\ z_2 \end{bmatrix} \tag{5}$$

and the MaxEnt policy transition matrix as

$$\pi = \begin{bmatrix} P_1 & * \\ 0 & P_2 \end{bmatrix}$$

where $P_2 = [1]$ corresponds to the self-absorbing edge at the destination, and $P_1$ corresponds to all other nodes.

The MaxEnt backward pass is unnormalized power iteration. After $k$ iterations, $z^{(k)} = A^k \mathbb{I}_{s_d}$ where $\mathbb{I}_{s_d}$ is the one-hot vector at the destination node. This is exactly (unnormalized) power iteration.

Likewise, the forward pass is a matrix geometric series. The *fundamental matrix* $S$ (J. H. Hubbard and B. B. Hubbard, 2003) is defined as

$$S = I + P_1 + P_1^2 + \cdots$$
$$= (I - P_1)^{-1} \tag{6}$$

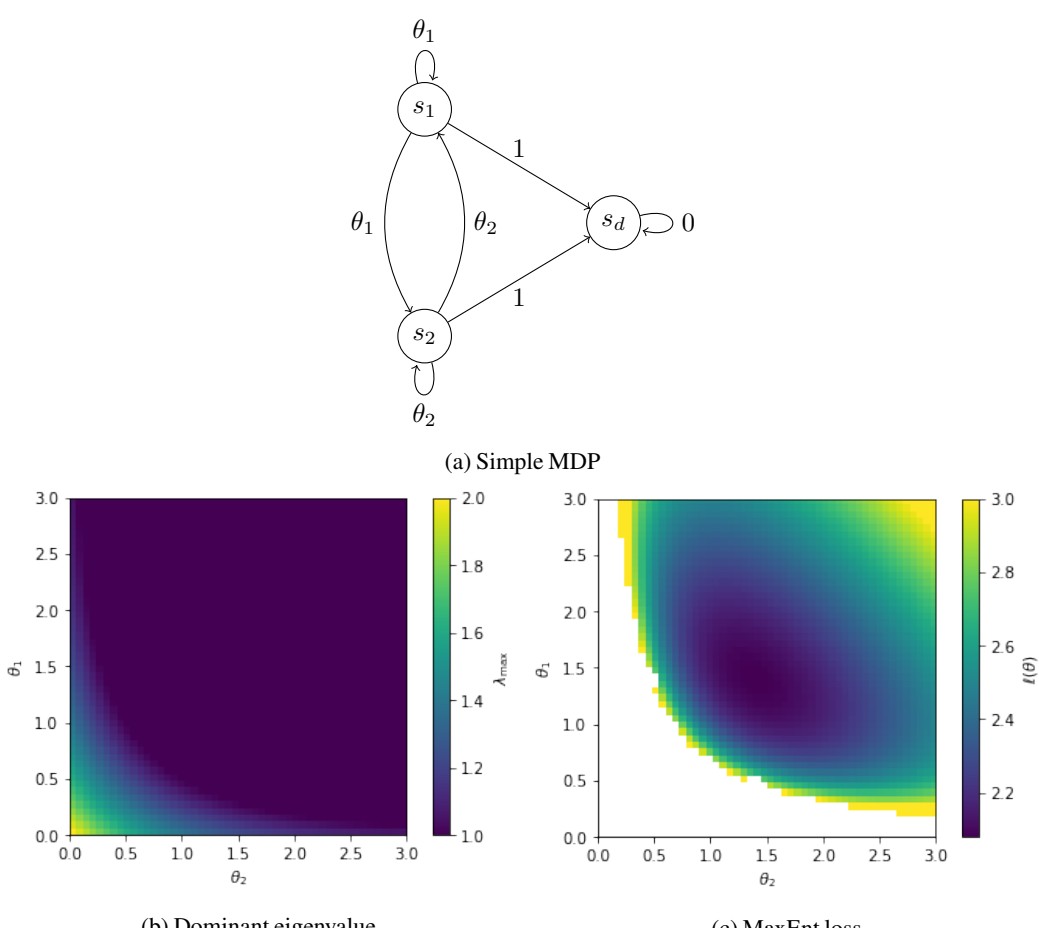

(a) Simple MDP

(b) Dominant eigenvalue

(c) MaxEnt loss

Figure 11: Demonstration of our claims on a simple MDP. (a) The deterministic MDP with absorbing destination node $s_d$, and parameterized edge costs (i.e. negative reward) $\theta_1$ and $\theta_2$. (b) The dominant eigenvalue of the graph as a function of $\theta$. Note the set of parameters $\Theta = \{\theta : \lambda_{\max}(\theta) = 1\}$ is convex. (c) The corresponding MaxEnt loss (for a dataset with two demonstration paths $s_1 \to s_2 \to s_d$ and $s_2 \to s_1 \to s_d$). As explained by our theory, the loss is finite and convex when $\lambda_{\max} = 1$ and infinite when $\lambda_{\max} > 1$.

Then the forward pass is simply $\mathbb{I}_{s_o}^\intercal S$, where $\mathbb{I}_{s_o}$ is the one-hot vector at the origin. If $P_1$ is sub-stochastic, then the series converges and $(I - P_1)$ is nonsingular. With these preliminaries, we now proceed to our main results.

## B.2 MAIN RESULTS

**Theorem B.1.** $\ell(\theta) < \infty$ *iff A has a dominant eigenvalue of 1.*

*Proof.* The spectrum of $A$, denoted $\sigma(A)$, is the union of $\sigma(B_1)$ and $\sigma(B_2)$, including multiplicities.[6] Thus, there are three possible cases:

1. **Case 1**: $\lambda_{\max}(B_1) < 1$. In this case, there exists a dominant eigenvector with eigenvalue 1 from $B_2$. This eigenvector must satisfy $z_2 > 0$ (proof by contradiction: assume $z_2 = 0$, then $B_1 z_1 = z_1$, which contradicts $\lambda_{\max}(B_1) < 1$). Since $z_2 > 0$, $P_1$ is substochastic (there is some probability of transitioning to the destination node), the matrix geometric series $(I + P_1 + P_1^2 + \cdots)$ converges, and the loss is finite.

---

[6]by 8.3.P8 in Horn and Johnson (2012)

2. **Case 2**: $\lambda_{\max}(B_1) > 1$. There exists a dominant eigenvector with eigenvalue $\lambda_{\max}(B_1)$. Then by Equation 5, $z_2$ must equal zero. By the MaxEnt policy definition, the policy will never transition to the destination ($\pi_{iN} = 0 \; \forall i \neq N$), the forward pass geometric series diverges, and thus $\ell(\theta) = \infty$.

3. **Case 3**: $\lambda_{\max}(B_1) = 1$. This determines whether the set is open or closed. We do not address this question in this paper, although we strongly suspect the set is open.

In both Cases 1 and 2, there exists a dominant eigenvector. By standard power iteration analysis, the MaxEnt backward pass is guaranteed to converge to this dominant eigenvector. The MaxEnt loss $\ell(\theta)$ is finite in Case 1 when $A$ has a dominant eigenvalue of 1, and infinite in Case 2. $\qquad\square$

Computing the dominant eigenvalue may be expensive. Fortunately, it is often possible to cheaply verify Case 1 holds by use of the following simple inequalities[7]

$$\lambda_{\max}(B_1) \leq \max_i r_i \sum_j B_{1,ij} r_j \leq \max_i r_i$$

$$\lambda_{\max}(B_1) \leq \max_j c_j \sum_i B_{1,ij} c_i \leq \max_j c_j$$

where $r_i = \sum_j B_{1,ij}$ and $c_j = \sum_i B_{1,ij}$ are the row and column sums, respectively.

**Theorem B.2.** $\Theta = \{\theta : \ell(\theta) < \infty\}$ *is an open convex set.*

*Proof.* From the proof of Theorem B.1, we have the equivalence

$$\Theta = \{\theta : \lambda_{\max}(B_1) < 1\}. \tag{7}$$

Let

$$\lambda_{\max}(B_1) = h(g(f(\theta)))$$

where

| Definition | Name | Relevant properties |
|---|---|---|
| $f(\theta) = X\theta$ | Edge rewards | Linear |
| $g(R) = e^R$ | Elementwise exponential | Convex, increasing |
| $h(B_1) = \lambda_{\max}(B_1)$ | Dominant eigenvalue | Log-convex, increasing |

The only non-obvious fact is that $h$ is convex and increasing. Log-convexity follows from Kingman (1961). It is increasing by the Perron-Frobenius theorem for non-negative irreducible matrices.

The proof follows standard convexity analysis (Boyd and Vandenberghe, 2004). The function $g \circ f$ is convex since $f$ is convex and $g$ is convex and increasing. The function $h \circ g \circ f$ is convex since $g \circ f$ is convex and $h$ is convex and increasing. Finally, since $\lambda_{\max}(B_1) = h \circ g \circ f$ is convex in $\theta$, the sublevel set in Equation 7 is also convex. $\qquad\square$

**Theorem B.3.** *The error of the MaxEnt backward pass after $k$ iterations is*

$$||z^{(k)} - z|| = \mathcal{O}\left(\left|\frac{\lambda_2}{\lambda_1}\right|^k\right) \tag{8}$$

*where $\lambda_1$ is the dominant eigenvalue of $A$ ($\lambda_1 = 1$ in Case 1), and $\lambda_2 < \lambda_1$ is the second largest eigenvalue.*

*Proof.* This follows directly from standard power iteration analysis, e.g. Theorem 27.1 in Trefethen and Bau (1997). $\qquad\square$

---

[7] by Lemma 8.1.21 and 8.1.P7 in Horn and Johnson (2012).

### B.3 PROOF OF EQUATION 2

In Section 4, we claim our MaxEnt++ initialization is closer to the desired solution than the original MaxEnt initialization

$$\underbrace{\mathbb{I}_{s=s_d}}_{\text{MaxEnt initialization}} \leq \underbrace{\min_{\tau \in \mathcal{T}_{s,s_d}} e^{r(\tau)}}_{\text{MaxEnt++ initialization}} \leq \underbrace{\sum_{\tau \in \mathcal{T}_{s,s_d}} e^{r(\tau)} = v}_{\text{Solution}}$$

The MaxEnt initialization $\mathbb{I}_{s=s_d}$ is zero everywhere, except at $s_d$ where it equals one. The MaxEnt++ initialization $\min_{\tau \in \mathcal{T}_{s,s_d}} e^{r(\tau)}$ is non-negative (due to the exponent), and is equal to one at $s_d$ (since there is a single, zero-reward, self-absorbing edge at the destination). Thus, the Max-Ent++ initialization upper bounds the MaxEnt initialization. The solution $\sum_{\tau \in \mathcal{T}_{s,s_d}} e^{r(\tau)} = v$ upper bounds the MaxEnt++ initialization by the inequality $\min_x f(x) \leq \sum_x f(x)$, where $f(x) \geq 0 \ \forall x$.

Note the claim in Equation 2 is stronger than claiming a smaller vector distance.

## C BASELINE ALGORITHMS

In this section, we summarize the baseline IRL algorithms and provide reductions from RHIP.

### C.1 MAXENT++ AND MAXENT

MaxEnt (Ziebart et al., 2008) assumes a softmax distribution over trajectories.

$$\pi(\tau) \propto e^{r(\tau)} \tag{9}$$

$$\pi(a|s) \propto \sum_{\tau \in \mathcal{T}_{s,a}} e^{r(\tau)} \tag{10}$$

where $\mathcal{T}_{s,a}$ is the set of all trajectories which begin with state-action pair $(s,a)$. Computing the gradient of the log-likelihood of Equation 9 results in dynamic programming Algorithm 2.

---

**Algorithm 2** MaxEnt++ and MaxEnt

---

**Input:** Reward $r_\theta$, demonstration $\tau$ with origin $s_o$ and destination $s_d$
**Output:** Parameter update $\nabla_\theta f$

```
# Policy estimation
```
$v^{(0)}(s) \leftarrow \text{DIJKSTRA}(r_\theta, s, s_d)$     ▷ Equation 2. The original MaxEnt uses $\log \mathbb{I}[s=s_d]$
**for** $h = 1, 2, \dots$ **do**
 $Q^{(h)}(s,a) \leftarrow r_\theta(s,a) + v^{(h-1)}(s')$
 $v^{(h)}(s) \leftarrow \log \sum_a \exp Q^{(h)}(s,a)$
**end for**
$\pi(a|s) \leftarrow \frac{\exp Q^{(h)}(s,a)}{\sum_{a'} \exp Q^{(h)}(s,a')}$        ▷ Equation 9

```
# Roll-out policy
```
$\rho^\theta \leftarrow \text{Rollout}(\pi, s_o)$

$\nabla_\theta f \leftarrow \sum_{s,a} (\rho_\tau(s,a) - \rho^\theta(s,a)) \nabla_\theta r_\theta(s,a)$
**Return:** $\nabla_\theta f$

---

The reduction from RHIP to MaxEnt follows directly from Equation 3 with $H = \infty$. Recall $\pi_s$ is the MaxEnt++ policy after $H$ iterations, and let $\pi_\infty$ denote the fully converged MaxEnt++ policy

(equivalent to the fully converged MaxEnt policy)

$$\pi(a|s) \propto \sum_{\tau \in \mathcal{T}_{s,a}} \pi_d(\tau_{H+1}) \prod_{h=1}^{H} \pi_s(a_h|s_h) \qquad \text{RHIP}$$

$$= \sum_{\tau \in \mathcal{T}_{s,a}} \prod_{h=1}^{\infty} \pi_\infty(a_h|s_h)$$

$$= \sum_{\tau \in \mathcal{T}_{s,a}} \pi_\infty(\tau) \qquad \text{MaxEnt++ and MaxEnt}$$

The careful reader will notice a subtle distinction between Algorithm 1 (RHIP) and Algorithm 2 (MaxEnt). The former computes the gradient via Equation 10, whereas the latter computes the gradient via Equation 9. The resulting gradients are identical, however, the latter results in a simpler algorithm. Mechanically, Algorithm 1 with $H=\infty$ reduces to the simpler Algorithm 2 via a telescoping series.

## C.2 BAYESIAN IRL

Bayesian IRL (Ramachandran and Amir, 2007) assumes the policy follows the form

$$\pi(a|s) \propto e^{Q^*(s,a)} \qquad (11)$$

where $Q^*(s,a)$ is the Q-function of the optimal policy. In our setting, $\pi_d$ is the optimal policy, and thus $Q^*(s,a)$ is the reward of taking $(s,a)$ and then following $\pi_d$, i.e. $Q^{(0)}$ in Algorithm 1. Computing the gradient of the log-likelihood of Equation 11 results in Algorithm 3

---

**Algorithm 3** Bayesian IRL

---

**Input:** Reward $r_\theta$, demonstration $\tau$ with origin $s_o$ and destination $s_d$
**Output:** Parameter update $\nabla_\theta f$

```
# Backup policy
```
$v(s) \leftarrow \text{DIJKSTRA}(r_\theta, s, s_d)$
$Q(s,a) \leftarrow r_\theta(s,a) + v(s')$
$\pi(a|s) \leftarrow \frac{\exp Q(s,a)}{\sum_{a'} \exp Q(s,a')}$               ▷ Equation 11

```
# Roll-out policy
```
$\rho^\theta \leftarrow \text{Rollout}(\pi, \rho_\tau^s)$
$\rho^* \leftarrow \text{Rollout}(\pi_{2:\infty}, \rho_{\tau_{2:\infty}}^s) + \rho_\tau^{sa}$

$\nabla_\theta f \leftarrow \sum_{s,a} (\rho^*(s,a) - \rho^\theta(s,a)) \nabla_\theta r_\theta(s,a)$
**Return:** $\nabla_\theta f$

---

The reduction from RHIP to BIRL directly follows from Equation 3 with $H=1$.

$$\pi(a|s) \propto \sum_{\tau \in \mathcal{T}_{s,a}} \pi_d(\tau_{H+1}) \prod_{h=1}^{H} \pi_s(a_h|s_h) \qquad \text{RHIP}$$

$$= \sum_{\tau \in \mathcal{T}_{s,a}} \pi_d(\tau_{H+1}) \pi_1(a_h|s_h)$$

$$= \sum_{\tau \in \mathcal{T}_{H,s,a}} \pi_1(a_h|s_h)$$

$$= \sum_{\tau \in \mathcal{T}_{H,s,a}} \pi_1(a_h|s_h)$$

$$= e^{Q^{(0)}(s,a)} = e^{Q^*(s,a)} \qquad \text{BIRL}$$

where $\mathcal{T}_{H,s,a}$ is the set of all paths which begin with a state-action pair $(s,a)$ and deterministically follow the highest reward path after horizon $H$.

## C.3 MAX MARGIN PLANNING

Max Margin Planning (Ratliff et al., 2006; Ratliff et al., 2009) assumes a loss $\ell$ of the form

$$\ell(\theta) = \underbrace{\sum_{s,a} \rho_\tau^{sa}(s,a) r_\theta(s,a)}_{\text{Demonstration path reward}} - \underbrace{\max_{\tau' \in \mathcal{T}} \sum_{s,a} \rho_{\tau'}^{sa}(s,a)(r_\theta(s,a) + m_\tau(s,a))}_{\text{Margin-augmented highest reward path}} \tag{12}$$

$$= \sum_{s,a}(\rho_\tau(s,a) - \rho_{\tau_{\text{sp}}}(s,a)) r_\theta(s,a)$$

where $\mathcal{T}$ is the set of all paths from $s_o$ to $s_d$, $m_\tau$ is the margin term, and $\tau_{\text{sp}}$ is the margin-augmented highest reward path, i.e. $\tau_{\text{sp}} = \text{DIJKSTRA}(r_\theta + m_\tau, s_o, s_d)$. The resulting MMP algorithm is shown in Algorithm 4.

---

**Algorithm 4** Max Margin Planning (MMP)

---

**Input:** Reward $r_\theta$, demonstration $\tau$ with origin $s_o$ and destination $s_d$
**Output:** Parameter update $\nabla_\theta f$

```
# Estimate and roll-out policy
```
$\rho_{\tau_{\text{sp}}} \leftarrow \text{DIJKSTRA}(r_\theta + m_\tau, s_o, s_d)$        ▷ Equation 12

$\nabla_\theta f \leftarrow \sum_{s,a}(\rho_\tau(s,a) - \rho_{\tau_{\text{sp}}}(s,a)) \nabla_\theta r_\theta(s,a)$
**Return:** $\nabla_\theta f$

---

The reduction from RHIP to MMP is possible by directly examining Algorithm 1 with $H = 0$ and absorbing the margin term into the reward function, i.e. $r'(\theta) = r(\theta) + m_\tau$. With $H = 0$, $\pi = \pi_d$ in Algorithm 1. Beginning with the $\rho^* - \rho^\theta$ term in Algorithm 1

$$\rho^* - \rho^\theta = \text{Rollout}(\pi_{2:\infty}, \rho_{\tau_{2:\infty}}^s) + \rho_\tau^{sa} - \text{Rollout}(\pi, \rho_\tau^s) \qquad \text{RHIP}$$
$$= \text{Rollout}(\pi_d, \rho_{\tau_{2:\infty}}^s) + \rho_\tau^{sa} - \text{Rollout}(\pi_d, \rho_\tau^s) \qquad H = 0$$
$$= \rho_\tau^{sa} - \text{Rollout}(\pi_d, s_o) \qquad \text{Telescoping series}$$
$$= \rho_\tau^{sa} - \rho_{\tau_{\text{sp}}}^{sa} \qquad \text{MMP}$$

# D EXPERIMENTS

## D.1 DATASET

The demonstration dataset contains 110M and 10M training and validation samples, respectively. Descriptive statistics of these routes are provided below.

| Travel model | Distance (km) | Duration (min) | Road segments per demonstration |
|---|---|---|---|
| Drive | 9.7 | 13.3 | 99.5 |
| Two-wheeler | 3.0 | 8.3 | 47.4 |

Table 3: Dataset summary statistics

GPS samples are matched to the discrete road graph using a hidden Markov model. For data cleaning, we relied on several experts who had at least 5 years of experience in curating routing databases. This is similar to data cleaning processes followed in prior work, for example in Derrow-Pinion et al. (2021).

## D.2 HYPERPARAMETERS

The full hyperparameter sweep used in Table 1 (excluding global result in bottom row) is provided in Appendix D.2.

| Category | Hyperparameter | Values | Applicable methods |
|---|---|---|---|
| Policy | Softmax temperature | 10, 20, 30 | MaxEnt, BIRL |
| | Horizon | 2, 10, 100 | RHIP |
| | Margin | 0.1, 0.2, 0.3 | MMP |
| | Fixed bias | Margin+0.001 | MMP |
| Reward model | Hidden layer width | 18 | DNN |
| | Hidden layer depth | 2 | DNN |
| | Initialization | $(1 \pm .02)*$ETA+penalties | Linear, DNN |
| | Weight initialization | 0 | SparseLin |
| | L1 regularization | 1e-7 | SparseLin |
| Optimizer | SGD learning rate | 0.05, 0.01 | Linear, DNN |
| | Adam learning rate | 1e-5, 1e-6 | SparseLin |
| | Adam $\beta_1$ | 0.99 | SparseLin |
| | Adam $\beta_2$ | .999 | SparseLin |
| | Adam $\epsilon$ | 1e-7 | SparseLin |
| | LR warmup steps | 100 | All |
| Training | Steps per epoch | 100 | All |
| | Batch size | 8 | All |
| | Epochs | 200, 400 | All |
| Data compression | Max out degree | 3 | All |

Table 4: Hyperparameters.

The worldwide 360M parameter RHIP model (bottom row of Table 1) uses the same parameters as Appendix D.2, except that we only swept over the horizon parameter in $H \in \{10, 100\}$ and the SGD learning rate, for efficiency. Temperature was fixed at 30 for linear and 10 for sparse. Adam learning rate was fixed at 0.05.

### D.3 STATISTICAL SIGNIFICANCE ANALYSIS

The $p$-values for Table 1 are computed as follows:

**Accuracy**  We used a two-sided difference of proportions test (*Difference of Proportion Hypothesis Test* 2009). Our validation set in the experimental region contains $n = 360,000$ samples. RHIP outperforms the next-best drive and two-wheeler policies by .0023 ($p$-value=.051) and .0018 ($p$-value=.122), respectively. Note that since policies share a validation set, the routes on which they have a perfect match are not independent but likely positively correlated, making these estimates conservative.

**IoU**  We used a Hoeffding bound to construct a confidence interval for the difference of IoU scores (Wasserman, 2004). We find RHIP does not provide a statistically significant improvement.

## D.4 ADDITIONAL RESULTS



**Preferred route**     **Bad detour**



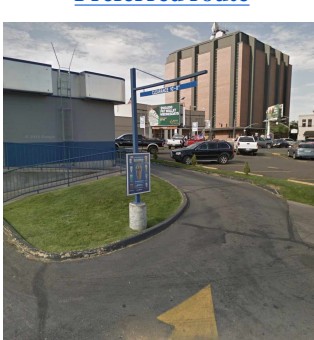 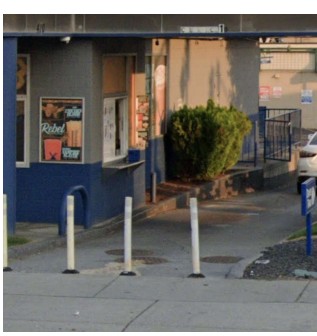

(a) **Spokane** A road is incorrectly marked as drivable. The correct route takes users through the designated drive-through, as desired. The sparse model learns to correct the data error with a large negative reward on the flex-posts segment.

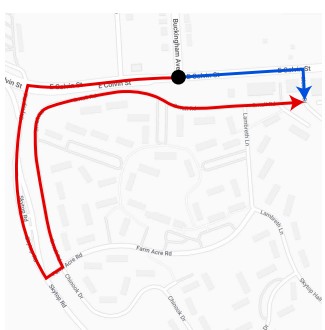 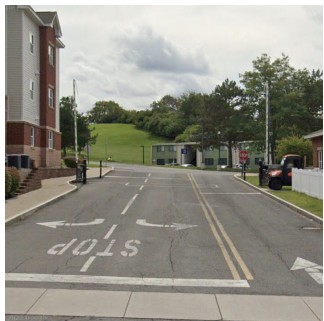 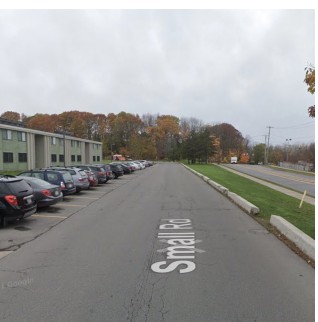

(b) **Syracuse** Similar to Figure 4, except this road segment is *occasionally* closed. The longer detour follows several back roads and parking lots.

Figure 12: Examples of the 360M parameter sparse model finding and correcting data quality errors. Locations were selected based on the largest sparse reward magnitudes in the respective regions.

