# OpenReview forum: "Massively Scalable Inverse Reinforcement Learning in Google Maps"
_ICLR.cc/2024/Conference — ICLR 2024 spotlight_

### Official Review · Reviewer_2Fnw · 2023-10-31

**Soundness:** 3 good
**Presentation:** 3 good
**Contribution:** 2 fair
**Rating:** 6
**Confidence:** 3

**Summary:**

This paper addresses the problem of route optimization. Given a set of demonstrations of chosen navigation routes that optimize a set of unknown preferences (e.g., concerning distance, traffic, pollution), the goal is to learn a model such that suitable routes can be suggested for (possibly unseen) route-destination pairs. The authors address this problem with inverse reinforcement learning, in which the goal is to learn the reward function underlying these preferences. Equipped with the reward function, routes can be suggested e.g. via finding the highest cumulative reward path between the source and destination.

The authors present a set of improvements over standard IRL algorithms, concerning an improved initialization of the MaxEnt algorithm,  learning separate reward functions per geographical region, and trading off between expensive stochastic rollouts and cheaper deterministic planners. The method is evaluated on a global dataset of routes in several cities, showing that the method compares favorably with other IRL algorithms.

## Post-response update
I am updating the score to 6 as a result of the discussion. I think the benefits of publishing the findings of this work outweigh the shortcomings.

**Strengths:**

**S1**. The work successfully scales IRL to a large, real-world setting, indeed representing (to the best of my knowledge) the largest-scale evaluation of IRL.

**S2**. Furthermore, it provides an interesting perspective on the inherent challenges of global route optimization, for example regarding the "locality" of the learned policies, suggesting individuals navigate differently in different cities. This may have wider implications in other domains e.g. transportation science, neuroscience.

**Weaknesses:**

**W1**. Methodological contributions: with the exception of the MaxEnt initialization findings, I am unsure of the value of the methodological developments. The geographical split into multiple experts and the graph compression are, in my opinion, both straightforward. I think simplicity is desirable, but the contribution is oversold.

**W2**. Generalizability and reproducibility: given the repeated nods to engineering and deployment constraints, how generalizable and reproducible are the results? How many organizations face global scale routing optimization? While the achieved improvements are definitely impressive in terms of e.g. customer satisfaction, the contribution to the scientific community is not clear-cut, especially given that code and data (I assume) will not be released. Reproducibility and code / data availability are not even mentioned in passing.

**Questions:**

Please see W1/W2 above. In terms of additional comments:

**C1**. The style of Figure 1 and Figure 2 is by now instantly recognizable and, in my opinion, represents a breach of anonymity.

**C2**. The wording "largest published benchmark of IRL algorithms [...]" (abstract, p.2, p.9) is misleading. I assume that the authors do not intend to publish the actual benchmark (e.g., data and evaluation metrics), but solely the results of this evaluation. This should be revised.

**C3**. Typos: "rouute" (Footnote 1)

---

> ### Author Response · Authors · 2023-11-13
> **Response to reviewer**
>
> Thank you for taking the time to read and review our paper!
>
> Please see our responses below. We've already uploaded some of the changes requested by the reviewers, and will incorporate all of them in the final version and credit anonymous reviewers in the acknowledgements. If you have any additional suggestions, we'd be happy to discuss them as well.
>
> * **Re: I am unsure of the value of the methodological developments** Some of our contributions are straightforward (e.g. geographic sharding) while others are much less obvious. For example, neither the MaxEnt++ initialization nor the RHIP methodological advancements were realized during the 15 years since the original MaxEnt paper was published. We believe those methodological developments, combined with a large-scale empirical study, new theoretical analyses that explains training instabilities in MaxEnt, and presentation of negative results represents a high-value publication for the inverse RL research community.
>
> * **Re: how generalizable and reproducible are the results?** In Figure 6 we studied one aspect of generalizability, namely the ability of reward models to generalize across geographic regions. The p-value analysis in Table 1 and Appendix D.3 provides strong statistical guarantees on our claims. Not shown in this paper, we also studied the generalizability of our results _across time_, and only noted a minor drop in performance. We'd be happy to discuss other dimensions for testing the generalizability of the results. [For reproducibility concerns, see below discussion on data availability]
>
> * **Re:  How many organizations face global scale routing optimization?** We're aware of a few organizations, plus a larger number of open-source projects for global routing based on OpenStreetMaps [1]. Perhaps more importantly, we want to call out Reviewer 5CwT's comment:
> > [Note to ACs and other reviewers]: Although the proposed method is framed for discrete MDP-based route optimization, note that there are several ways to generalize this framework to other interesting problem settings quite trivially, (see e.g. [A]) - as such, the findings here are actually quite broadly applicable.
>
>     We chose this setting primarily because it's the largest scale application we could find to test our methods on.
>
> * **Re: data and code availability** Due to strict user privacy requirements, we believe it's unlikely a routing dataset of the scope used in this paper could be publicly released in the foreseeable future. We strongly support open-sourcing experiments when legally and ethically permissible.
>
> * **style of Figure 1 and Figure 2 is by now instantly recognizable and, in my opinion, represents a breach of anonymity** We based the style on figures from other papers that we liked (e.g. Figure 1 in [2]), and created the graphics using the publicly available Adobe Illustrator with open-source fonts and graphics from Adobe Stock. Could you provide some more details to how this violates ICLR's anonymity requirements?
>
> * **Re: "largest published benchmark"** Perhaps "largest published empirical study" is more precise? This could also be replaced with "empirical analysis" or "experiment".
>
> * **Re: Typo in footnote 1** Fixed! Thanks.
>
> ### References
> [1] https://wiki.openstreetmap.org/wiki/Routing#Developers
>
> [2] Jumper, John, et al. "Highly accurate protein structure prediction with AlphaFold." Nature 596.7873 (2021): 583-589.

---

> > ### Comment · Reviewer_2Fnw · 2023-11-17
> > **Response to authors**
> >
> > Thanks for engaging with my points!
> >
> > **On wider applicability and generalizability**: the examples provided by yourselves and the other reviewers are quite convincing, and I'd recommend emphasizing this aspect in the paper.
> >
> > **On data and code availability**: I am not recommending to prevent publication based on data unavailability (as a previous revision of this comment was suggesting). However, there is a general expectation in the scientific community (and rightly so) that efforts should be made to increase the reproducibility of works, which needs to be balanced against other considerations such as privacy. I do not think releasing an anonymized subset of this data (e.g., by considering journeys without their beginning and end segments, or only those at a very macroscopic level) is impossible, but there are certainly other considerations and implications. Regarding code, I think at least a reference implementation should be made available, given the contribution of the work rests on making IRL practical. In any case, I think that both data and code availability should be addresed in the paper, as they currently are not mentioned at all.
> >
> > **On anonymity**: My intention with this comment was to highlight it as a *possible* breach of anonymity that, while not explicitly covered by guidelines, might unintentionally influence the review process. To clarify, I am not suggesting that the paper be disqualified based on this. Given (1) there are very few organizations that have this type of data readily accessible and (2) the visual style of the figures including fonts and choice of colours adheres very closely to those used by one such organization in their public-facing communications, making this intuitive inference is not a leap (even if it may be incorrect).
> >
> > I am updating the score to 6 as a result of the discussion. I think the benefits of publishing the findings of this work outweigh the shortcomings.

---

> > > ### Author Response · Authors · 2023-11-18
> > > **Thank you**
> > >
> > > Thanks for the productive discussion! We'll credit the anonymous reviewers in the final acknowledgements for their suggestions and helpful feedback.
> > >
> > > **On wider applicability and generalizability** Will do!
> > >
> > > **On data and code availability** We'll address these points in the paper, and release the code if deemed feasible.
> > >
> > > **On anonymity** Point taken, we can use alternative figure styles for future submissions.

---

### Official Review · Reviewer_Ucz3 · 2023-11-01

**Soundness:** 3 good
**Presentation:** 3 good
**Contribution:** 3 good
**Rating:** 6
**Confidence:** 4

**Summary:**

The paper proposes MaxEnt++, an adaptation of the classical MaxEnt algorithm, to handle very large route instances with hundreds of millions of states and demonstration trajectories. Their techniques include MaxEnt++, a MaxEnt algorithm with a DIJKSTRA component, a new policy formulation that the authors call Receding Horizon Inverse Planning (RHIP), and a graph compression technique to reduce memory usage. The algorithm was then tested with a routing dataset of 200M states, showing some improvements compared to the standard MaxEnt and other baselines.

**Strengths:**

The paper addresses an interesting problem. Learning with very large-scale routing datasets would have significant applications in modern transportation systems. The techniques used in the paper (except for MaxEnt, as I will discuss in the Weaknesses) are sound and relevant. The algorithm seems to work well (but again, the experiments lack comparisons with more scalable IRL algorithms, as I will discuss later).

**Weaknesses:**

My biggest concern is that the paper primarily revolves around MaxEnt, which was developed about 15 years ago and is now very outdated. In the introduction, the authors state that MaxEnt is limited in its scalability, which is true. Recent literature on IRL has introduced many advanced algorithms to address this issue. For instance, Adversarial IRL [1] and IQ-Learn [2], value DICE [3] are well-known recent IRL algorithms that are much more scalable. Therefore, it is crucial to focus on these algorithms instead of the outdated MaxEnt.

[1] Fu, Justin, Katie Luo, and Sergey Levine. "Learning robust rewards with adversarial inverse reinforcement learning." ICLR 2018.

[2] Garg, Divyansh, Shuvam Chakraborty, Chris Cundy, Jiaming Song, and Stefano Ermon. "IQ-Learn: Inverse Soft-Q Learning for Imitation." Advances in Neural Information Processing Systems 34 (2021): 4028-4039.

[3] Kostrikov, Ilya, Ofir Nachum, and Jonathan Tompson. "Imitation learning via off-policy distribution matching." ICLR 2019

I notice that the related work section exclusively references older papers and appears to be outdated. It would be beneficial for the authors to give greater consideration to more recent developments in the field of IRL/imitation learning.

This should be noted that the routing task is deterministic, so both online and offline IRL/imitation learning algorithm can be applied. The authors should look at relevant works and make a complete comparison.

**Questions:**

I do not have many questions about the current work, as the current contributions are not convincing, and the paper clearly needs much more work to reach a publishable level.

# Post-rebuttal:

I have increased my score to 6. There are some remaining concerns but I think the paper has some good merits.

---

> ### Author Response · Authors · 2023-11-13
> **Response to reviewer**
>
> Thank you for taking the time to read and review our paper!
>
> Please see our responses below. We've already uploaded some of the changes requested by the reviewers, and will incorporate all of them in the final version and credit anonymous reviewers in the acknowledgements. If you have any additional suggestions, we'd be happy to discuss them as well.
>
> * **Re: IQ-Learn, Adversarial IRL and ValueDICE** We've used these algorithms in other domains, but there is a subtle aspect of the discrete routing problem that makes them poorly suited for this setting (a point we've discussed with some of the original authors of these papers as well). Specifically, the _goal conditioning requirement_ specifies that each destination $s_d$ induces a slightly different MDP (where the destination state has been modified to be self-absorbing with zero-reward). In the tabular setting, the number of reward parameters is $\mathcal{O}(SA)$ even when conditioning on $s_d$. This is in contrast to approaches that explicitly learn a Q-function (e.g. IQ-Learn) or value function (e.g. ValueDICE), where the number of required parameters increases from $\mathcal{O}(SA)$ to $\mathcal{O}(S^2A)$ and $\mathcal{O}(S)$ to $\mathcal{O}(S^2)$, respectively. In other words, by directly learning rewards, we're able to trivially transfer across MDPs with different destination states $s_d$, unlike approaches that learn a policy, Q-function or value function.
>
>     AIRL samples trajectories from the current policy, trains a binary trajectory discriminator, performs a simple transformation to compute rewards from the logistic discriminator probabilities, and then updates the policy with any policy optimization method. In discrete settings, it appears the original paper authors actually use the MaxEnt policy for the final step (see Section 7.1 of [2]). We could follow the same approach – one key difference between the resulting algorithm and the techniques we consider is that instead of exactly computing the policy's stationary distribution, the AIRL technique would sample trajectories. Although sampling trajectories is required in continuous settings, it will almost certainly have higher variance compared to exactly computing the policy's stationary distribution (which we're able to do in the discrete setting).
>
> * **Re: MaxEnt is outdated** Recent works (including modern methods such as IQ-Learn!) build off MaxEnt. Our work similarly builds off and advances MaxEnt, and also incorporates modern optimization techniques and function approximators. The original MaxEnt is simply one of our baselines, and you can see in Table 1 that newer baselines (e.g. [1]) outperform the original paper, as expected.
>
> * **Re: Online and offline IRL/imitation learning algorithms can be applied** Is there a specific method that you have in mind? We'd be more than happy to discuss alternatives that can be applied in this setting.
>
> ### References
>
> [1] Wulfmeier, Markus, Peter Ondruska, and Ingmar Posner. "Maximum entropy deep inverse reinforcement learning." arXiv:1507.04888 (2015).
>
> [2] Fu, Justin, Katie Luo, and Sergey Levine. "Learning robust rewards with adversarial inverse reinforcement learning." International Conference on Learning Representations (2018).

---

> > ### Comment · Reviewer_Ucz3 · 2023-11-13
> > **A comparison with other baselines  is necessary.**
> >
> > I thank the authors for the responses. I, however, find them not convincing:
> >
> > - Other algorithms are purely suited for route optimization settings: I do not find this point convincing without any experiments provided.
> > - "goal conditioning requirement": I believe other algorithms such as IQ-learn can be adapted quite easily to handle this. For example, for IQ-learn, you just need to define and learn destination-dependent Q and V functions. Moreover, as far as I know, other modern IRL algorithms can handle discrete domains well.
> > - "Number of parameters increases": it is not necessarily the case. You can keep the same parameters across destinations, similarly to our approach.
> > - The other recent algorithms (e.g., AIRL) can be adapted in the same ways.
> > - ValueDICE is an offline imitation algorithm. I just meant the recent literature on imitation learning and IRL has introduced quite a few good algorithms (including the IQ-learn and DICE ones), so it is necessary to make comparisons against them to make the results convincing.

---

> ### Author Response · Authors · 2023-11-13
> **Destination-dependent Q and V functions**
>
> Thanks for the fast response!
>
> It seems the main point of contention is on the choice of baselines, and in particular the effects of adding destination-dependent Q and V functions to methods such as IQ-Learn and ValueDICE.
>
> * **Re: you just need to define and learn destination-dependent Q** Exactly, destination-dependent IQ-Learn would learn $Q(s, a | s_d)$. In the tabular setting, this function requires a parameter for every state-action-destination tuple, for a total of $\mathcal{O}(S^2A)$ parameters.
> * **Re: Number of parameters** If we instead "keep the same parameters across destinations" _in the tabular setting_, then won't this Q-function no longer be destination-dependent? Since the parameters are identical for each destination, then $Q(s, a | s_d) = Q(s, a) \forall s_d$ holds. In the tabular setting, it seems impossible to change from $Q(s, a)$ to $Q(s, a | s_d)$ without increasing the number of parameters. The analysis in the non-tabular setting is less straightforward, but the intuition is the same: IQ-Learn has to learn both the rewards and how to reach the destination, whereas reward-based techniques (MaxEnt, MMP, BIRL) only have to learn the former.
> * **Re: I do not find this point convincing without any experiments** We initially considered these baselines (IQ-Learn, ValueDICE) for our experiments, but ultimately chose other baselines that we believed were stronger due to the above concerns. We include a large number of baselines in Table 1 that aren't impacted by destination dependency, and we believe follow reasonable selection criteria.
>
> We also want to emphasize that for online routing requests, we require learned rewards to take advantage the reward pre-computation, contraction hierarchy, and fast graph search tricks described at the end of Section 3. These rewards could be estimated from IQ-Learn's Q-function, although this creates additional complexity, whereas we follow a more direct approach by directly learning the rewards.

---

> ### Comment · Reviewer_Ucz3 · 2023-11-14
> **Thanks for the responses**
>
> Thanks the authors for the responses:
>  -  $\mathcal{O}(S^2 A)$  parameters:  Have you attempted to train with this setting. I do not think a large number of parameters is a big issue in modern DL. If it is not possible to learn with this large number of parameters, then you can provide a comparison using a smaller network setting.
> - Will you make the code and data publicly available?

---

> ### Author Response · Authors · 2023-11-14
> **$O(S^2A)$ and smaller settings**
>
> We appreciate your continued attention to discussing our paper!
>
> * **Re: $\mathcal{O}(S^2A)$ parameter setting** The worldwide road graph contains $S \approx 200M$ states and $A < 10$ actions, implying that IRL methods such as destination-dependent IQ-Learn that scale $\mathcal{O}(S^2A)$ would require more than 4e16 parameters with these networks (thousands of times larger than GPT4). This scaling law holds with our best-performing reward models (DNN+SparseLin in Table 1), since they include a sparse reward parameter for every road in the graph, although not with some of the lower-quality reward networks (see discussion in next point).
>
> * **Re: Smaller settings** IQ-Learn and ValueDICE could be studied with smaller road graphs or reward models that exclude the empirically beneficial SparseLin features. Since the focus of our paper is on massively scalable IRL, we chose to only include baselines that we thought had a reasonable chance at success in the full worldwide destination-dependent setting. We think that studying methods such as IQ-Learn and ValueDICE in other settings is perhaps best left for a separate paper.
>
> * **Re: data availability** Due to strict user privacy requirements, we believe it's unlikely a routing dataset of the scope used in this paper could be publicly released in the foreseeable future. We strongly support open-sourcing experiments when legally and ethically permissible.
>
> * **Re: code availability** We're investigating the feasibility of releasing our code independent of the dataset.

---

> ### Comment · Reviewer_Ucz3 · 2023-11-17
> **Response to authors**
>
> I thank the authors for their responses. I am now more positive about the contributions of the work. I, however, still think that there would be ways to make IQ-learn or AIRL work in your context. More discussions would be appreciated.  Moreover, I am still concerned about the data and code availability because if you do not make them available, then no one can benchmark against your approach and it is impossible to validate your claims. I will increase my score to 6 as I think the paper has some good merits.

---

> > ### Author Response · Authors · 2023-11-18
> > **Thank you**
> >
> > Thanks for the productive discussion! We'll add those discussion points in the final version of the paper, and credit the anonymous reviewers for their suggestions and helpful feedback.

---

### Official Review · Reviewer_nMaV · 2023-11-01

**Soundness:** 3 good
**Presentation:** 3 good
**Contribution:** 3 good
**Rating:** 8
**Confidence:** 3

**Summary:**

This paper mainly focuses on scaling inverse reinforcement learning (IRL) for route optimization by learning the reward function from expert demonstrations. The application scenario is a popular route recommendation platform that should be able to generalize globally. Given a dataset of expert trajectories, in this approach, a reward function is learned from these demonstrations and this reward then guides an action selection policy from the start state to the destination.

Building on prior work in IRL, particularly MaxEnt IRL, the authors propose an initialization strategy that leads to faster convergence, called MaxEnt++. Next, they generalize these and other IRL algorithms in their proposed framework called RHIP (Receding Horizon Inverse Planning) that trades-off using an expensive stochastic policy upto a horizon H with a cheap deterministic planner afterwards. Additionally, a number of parallelized computation and graph compression techniques are implemented to further improve the scalability of their algorithm for the application setting. Experiments on held-out validation trajectories show the superior performance of their method compared to prior work in IRL for quality route recommendations.

**Strengths:**

1. The authors address a well-motivated and useful application to show the statistically significant gains obtained from scalable IRL in route recommendation. The techniques that worked for this task have been clearly explained, along with explanations and evidence for some techniques that didn't work.

2. The proposed method unifies several prior IRL algorithms through the RHIP framework for trading-off quality of route recommendation with convergence speed. This helps improve understanding of the similarities and differences in these approaches.

3. Several ablation studies have been performed for different graph compression techniques and reward modeling approaches that help establish the significance of the experimental results.

**Weaknesses:**

1. The experimental results are not from real-time execution of the proposed method and utilizes static features of the road network for route optimization. Incorporating dynamic features, for example varying traffic flow throughout a day, planned or unplanned diversions and road closures etc. would increase the difficulty of obtaining a scalable DP approach.

2. The reward function is learning a scalar value, whereas in the real world for applications like route optimization, it should intuitively be a multi-objective optimization problem. It is not immediately clear whether such possibilities would fit into the proposed algorithmic framework.

**Questions:**

1. The paper does not provide much details about the road graph. Would the authors be able to provide any intuition about the relation between the coarseness of the road network graph and the choice of H?

2. Fig 4 and 12 highlight an interesting outcome of the sparse reward modeling approach in correcting data quality errors. Is this a consistent observation across different geographical regions? Or is there any noticeable difference in the road graph network when this method of reward modeling demonstrates a particular benefit over others?

3. It is not quite clear what Fig. 7 is meant to convey. Could the authors explain more?

---

> ### Author Response · Authors · 2023-11-13
> **Response to reviewer**
>
> Thank you for taking the time to read and review our paper!
>
> Please see our responses below. We've already uploaded some of the changes requested by the reviewers, and will incorporate all of them in the final version and credit anonymous reviewers in the acknowledgements. If you have any additional suggestions, we'd be happy to discuss them as well.
>
> * **Re: Real-time features** Due to the format of our training examples (described shortly), using dynamic instead of static features would actually have no impact on training time or dataset disk space. Every example in our dataset consists of $(\tau, \mathcal{M}\_i)$ where $\tau$ is the demonstration trajectory and $\mathcal{M}\_i$ is the corresponding region MDP (with static features). Although this format uses significant disk space due to storing an MDP with every demonstration trajectory, it simplifies training since examples are fully self-contained and may be serially read from disk. Disk space is also relatively cheap compared to compute costs.
>
>     To switch from static to dynamic features, one could simply replace the static features in $\mathcal{M}\_i$ with a snapshot of the dynamic features at the time of the demonstration $\tau$. There would be zero change to the training pipeline – the lack of dynamic features is purely a limitation of the upstream logging infrastructure that creates $\mathcal{M}\_i$ and not a limitation of our machine learning setup.
>
> * **Re: Multi-objective optimization problem** Sorry, we're not sure we follow how this should be a multi-objective optimization problem, instead of a problem with scalar rewards. Do you have a reference? The existing work we're familiar with in this space uses scalar rewards [1, 2, 3, 4, 5]. Are you referring to adding user-dependent rewards, e.g. personalization [7]?
>
> * **Re: Road graph coarseness and choice of $H$** Broadly speaking, a coarser road graph will require a smaller choice of $H$. For example, if we split every road segment (node) into two equal shorter road segments, then one could double $H$ to get exactly equivalent training results. Note that with the 'merge' compression strategy, there is a single node between every feasible turn, meaning that the road graph is as coarse as possible without removing roads from the graph. As discussed in [6] and Section 6, we could prune additional nodes from the graph to make it even more coarse, although this creates significant engineering complexity.
>
>     We would be happy to discuss any additional details about the road graph that you're interested in.
>
> * **Re: Fig 4 and 12** At least anecdotally, we observed this behavior across all geographies that we manually inspected. We suspect it occurs more frequently in regions with a higher rate of data quality issues, although statistically verifying this claim is challenging without knowing the true issue rate (an unobserved quantity).
>
> * **Re: Fig 7** We meant for this figure to convey the fact that our method performs consistently across regions $\mathcal{M}\_1, \dotsc, \mathcal{M}\_m$ with respect to the size of the region, and that the total worldwide lift was not due to large gains in a small number of regions.
>
> ### References
>
> [1] Ziebart, Brian D., et al. "Maximum entropy inverse reinforcement learning." AAAI. Vol. 8. 2008.
>
> [2] Ho, Jonathan, and Stefano Ermon. "Generative adversarial imitation learning." Advances in Neural Information Processing Systems 29 (2016).
>
> [3] Finn, Chelsea, Sergey Levine, and Pieter Abbeel. "Guided cost learning: Deep inverse optimal control via policy optimization." International Conference on Machine Learning. PMLR, 2016.
>
> [4] Garg, Divyansh, et al. "IQ-Learn: Inverse soft-q learning for imitation." Advances in Neural Information Processing Systems 34 (2021).
>
> [5] Kostrikov, Ilya, Ofir Nachum, and Jonathan Tompson. "Imitation learning via off-policy distribution matching." International Conference on Learning Representations (2020).
>
> [6] Ziebart, Brian D. "Modeling purposeful adaptive behavior with the principle of maximum causal entropy." Carnegie Mellon University, 2010.
>
> [7] Nguyen, Quoc Phong, Bryan Kian Hsiang Low, and Patrick Jaillet. "Inverse reinforcement learning with locally consistent reward functions." Advances in neural information processing systems 28 (2015).

---

> > ### Comment · Reviewer_nMaV · 2023-11-21
> > **Response to authors**
> >
> > Thank you for your response and the clarifications.
> >
> > Re: Multi-objective optimization: Yes, I was referring to adding user-dependent rewards, similar to [7]. I would guess that for the type of application being studied in the paper, enabling personalization in route recommendation would be desirable in practice.
> >
> > I do not have any further comments on the paper, but as other reviewers have mentioned, I would also encourage possibly releasing the code and trying to maintain the anonymity of the submission. When a paper has already been widely publicized before submission for review (and I do understand that this is the current norm), I am not sure how helpful it will be to just use alternate figure styles in submissions.

---

> > > ### Author Response · Authors · 2023-11-21
> > > **Personalization**
> > >
> > > Thanks for the response! To follow-up on your final point:
> > >
> > > **Re: multi-objective optimization & personalization** Yes, rewards could be made user-dependent by adding user-specific features into each sample tuple $(\tau, \mathcal{M}\_i)$ (similar to switching from static to dynamic ETA features). From an ML standpoint, this would only require small changes to the reward model architecture to accommodate any new feature fields. We briefly discuss personalization as future work at the top of page 9, for legal and privacy reasons these user-specific features are currently unavailable to us.

---

### Official Review · Reviewer_5CwT · 2023-11-06

**Soundness:** 4 excellent
**Presentation:** 3 good
**Contribution:** 4 excellent
**Rating:** 10
**Confidence:** 4

**Summary:**

This paper considers the important problem of Inverse Reinforcement Learning for route optimization and planning, and specifically the practical limitations of existing methods when scaled to larger problems. The authors draw a unified theoretical perspective around Max Margin planning (Ratliff et al. 2006), the classic MaxEnt framework (Ziebart et al. 2008), and Bayesian IRL (Ramachandran and Amir 2007) which is helpful and insightful. Connections with graph theory methods lead to a novel IRL initialization and algorithm (MaxEnt++ and Receding Horizon Inverse Planning) which demonstrates significant improvements over other methods for large-scale problems. Several other graph optimization methods are presented which further allow scaling to global-scale routing problems.

The paper is well written, clear to read (despite covering a lot of theory and background), and the experimental evaluations are thorough and provide support for the claims.

I have a background in Inverse Reinforcement Learning theory, however have focused in other areas of computer science more recently, so may be out-of-touch with some recent literature results when performing this review. I have read the paper and appendices closely, however did not check the proofs carefully.

**Strengths:**

* A compelling problem
 * Real-world empirical experimental problem considered
 * The paper does a good job straddling both novel theory advancements, and practical and engineering advancements, but presents the findings appropriately for the ICLR audience.
 * The connections with graph theoretic results (App. A1, A2, and Theorem B3) are useful and insightful.
 * The paper and appendices include negative results, in addition to the main results - this is encouraging to see (more papers should do this).
 * [Note to ACs and other reviewers]: Although the proposed method is framed for discrete MDP-based route optimization, note that there are several ways to generalize this framework to other interesting problem settings quite trivially, (see e.g. [A]) - as such, the findings here are actually quite broadly applicable, as noted by the authors in Sec 6.

# References

 * [A] Byravan, Arunkumar, et al. "Graph-Based Inverse Optimal Control for Robot Manipulation." IJCAI. Vol. 15. 2015.

**Weaknesses:**

* The literature review is compact and the theory background provides a rapid but very nice summary of classical IRL results (in particular the unifying view of stochastic vs. deterministic policy trade-offs is helpful). One relevant piece of prior work that isn't mentioned however is the improved MaxEnt approach(es) by Snoswell et al. (e.g. [B, C]) - which address theoretical and empirical limitations with Ziebart's MaxEnt model, and are specifically applied to the problem of route optimization (albeit at city scale, not a global scale).

# References

 * [B] Snoswell, Aaron. "Modelling and explaining behaviour with Inverse Reinforcement Learning: Maximum Entropy and Multiple Intent methods." (2022).
 * [C] Snoswell, Aaron J., Surya PN Singh, and Nan Ye. "Revisiting maximum entropy inverse reinforcement learning: new perspectives and algorithms." 2020 IEEE Symposium Series on Computational Intelligence (SSCI). IEEE, 2020.

**Questions:**

# Questions and comments

 * Unclear notation - Paragraph 'Parallelism strategies' under Sec 4 defines the global reward based on the sharded MDP as $r(s,a) = r_{\theta_i}(s,a),(s,a) \in \mathcal{M}_i$. This notation isn't clear to me - is there a typo here? Is this mean to be a product over the sharded individual reward functions?

# Minor comments and grammatical points

 * Typo in footnote 1 - 'rouute'

---

> ### Author Response · Authors · 2023-11-13
> **Response to reviewer**
>
> Thank you for taking the time to read and review our paper!
>
> Please see our responses below. We've already uploaded some of the changes requested by the reviewers, and will incorporate all of them in the final version and credit anonymous reviewers in the acknowledgements. If you have any additional suggestions, we'd be happy to discuss them as well.
>
> * **Re: Snoswell et al.** Good catch, thanks. We've added these to our related work section.
>
> * **Re: Sec 4 parallelism strategies notation** [context] The global $\mathcal{M}$ is partitioned into $m$ disjoint MDPs $\mathcal{M}\_1, \dotsc, \mathcal{M}\_m$, where each state $s$ belongs to exactly one $\mathcal{M}\_i$. Each MDP $\mathcal{M}\_i$ has a corresponding learned reward $r\_{\theta\_i}$.
>
>    We meant for the notation in this section to express the fact that for state-action pair $(s, a)$ corresponding to $\mathcal{M}\_i$, the reward $r(s, a)$ is solely determined by region-specific expert $r\_{\theta\_i}$. In other words, the reward of each $(s, a)$ is determined by a single $r\_{\theta\_i}$. Perhaps a more clear notation here would be $r(s, a) = r\_{\theta\_i}(s, a)$ **where** $(s, a) \in \mathcal{M}\_i$? We're open to alternative suggestions.
>
> * **Re: Typo in footnote 1** Fixed! Thanks.

---

### Meta-Review · Area_Chair_ZaCf · 2023-12-03

**Metareview:**

The paper describes several important improves to the maximum entropy framework for inverse RL.   More specifically, the paper improves the initialization of the MaxEnt algorithm.  It also learns separate reward functions per geographical region, and trades off between expensive stochastic rollouts and cheaper deterministic planners. The method is evaluated on a global dataset of routes in several cities, showing that the method compares favorably with other IRL algorithms.  This advances the scalability of IRL techniques, while introducing the largest IRL benchmark to date.

**Justification For Why Not Higher Score:**

The advances appear to be somewhat tailored to the application of route optimization, however in the discussion with the reviewers there was a general agreement the proposed avdances should be applicable to other settings.

**Justification For Why Not Lower Score:**

The reviewers unanimously recommend recommendation.  There is a clear practical advance with a nice demonstration on a real world application domain.

---

### Decision · Program_Chairs · 2024-01-16

Accept (spotlight)